# Pharmacological inhibition of Mint3 attenuates tumour growth, metastasis, and endotoxic shock

Takeharu Sakamoto [1,2 ✉], Yuya Fukui[2,3], Yasumitsu Kondoh [4], Kaori Honda[4], Takeshi Shimizu[4], Toshiro Hara[5], Tetsuro Hayashi[6], Yurika Saitoh[7], Yoshinori Murakami [6], Jun-ichiro Inoue [3], Shuichi Kaneko [2], Hiroyuki Osada [4] & Motoharu Seiki[5]

Hypoxia-inducible factor-1 (HIF-1) plays essential roles in human diseases, though its central role in oxygen homoeostasis hinders the development of direct HIF-1-targeted pharmacological approaches. Here, we surveyed small-molecule compounds that efficiently inhibit the transcriptional activity of HIF-1 without affecting body homoeostasis. We focused on Mint3, which activates HIF-1 transcriptional activity in limited types of cells, such as cancer cells and macrophages, by suppressing the factor inhibiting HIF-1 (FIH-1). We identified naphtho-fluorescein, which inhibited the Mint3–FIH-1 interaction in vitro and suppressed Mint3-dependent HIF-1 activity and glycolysis in cancer cells and macrophages without evidence of cytotoxicity in vitro. In vivo naphthofluorescein administration suppressed tumour growth and metastasis without adverse effects, similar to the genetic depletion of Mint3. Naphthofluorescein attenuated inflammatory cytokine production and endotoxic shock in mice. Thus, Mint3 inhibitors may present a new targeted therapeutic option for cancer and inflammatory diseases by avoiding severe adverse effects.

[1] Department of Cancer Biology, Institute of Biomedical Science, Kansai Medical University, Shin-machi, Hirakata, Osaka, Japan. [2] Department of System Biology, Institute of Medical, Pharmaceutical and Health Sciences, Kanazawa University, Takaramachi, Kanazawa, Ishikawa, Japan. [3] Division of Cellular and Molecular Biology, The Institute of Medical Science, The University of Tokyo, Shirokanedai, Minato-ku, Tokyo, Japan. [4] Chemical Biology Research Group, RIKEN Center for Sustainable Resource Science, Wako, Saitama, Japan. [5] Division of Cancer Cell Research, Institute of Medical Science, The University of Tokyo, Shirokanedai, Minato-ku, Tokyo, Japan. [6] Division of Molecular Pathology, Institute of Medical Science, The University of Tokyo, Shirokanedai, Minato-ku, Tokyo, Japan. [7] Center for Medical Education, Teikyo University of Science, Senjusakuragi, Adachi-ku, Tokyo, Japan. ✉email: sakamott@hirakata.kmu.ac.jp

Oxygen is essential for eukaryotes, and cellular oxygen-sensing mechanisms play important roles in maintaining homoeostasis and in developing various diseases, such as cancer and inflammatory diseases. Among the cellular oxygen-sensing mechanisms, hypoxia-inducible factor-1 (HIF-1) is one of the most important transcriptional factors that promotes the expression of various genes related to metabolic pathways in glycolysis, angiogenesis, erythropoiesis, among others, in adaptive responses to hypoxic conditions[1–5]. HIF-1 is a heterodimeric transcription factor that comprises a regulatory α-subunit and a constitutive β-subunit. HIF-1α is negatively regulated by two types of hydroxylases, a HIF prolyl hydroxylase domain-containing protein 1−3 (PHD1−3) and the factor inhibiting HIF-1 (FIH-1), in an oxygen-dependent manner. PHD1−3 hydroxylates two proline residues of HIF-1α and, thereby, promotes proteasomal degradation of the HIF-1α protein. In turn, FIH-1 hydroxylates an asparagine residue of HIF-1α and, thereby, inactivates the transcriptional activity of HIF-1α[1,3,5]. Many studies have demonstrated that HIF-1 inhibition is effective for cancer therapy in various experimental models[6]; however, a specific HIF-1 inhibitor is not clinically available. HIF-1 is essential not only for cancer and inflammatory diseases but also for body homoeostasis; HIF-1α knockout in mice is embryonically lethal at day E11 due to cardiovascular malfunctions and neural tube defects, whereas HIF-1α conditional knockout mice show various defects in this regard[7–12]. These results indicate that HIF-1 inhibition itself can cause many adverse effects within the body. In particular, systemic HIF-1 inhibition would be contraindicated for individuals with heart failure because HIF-1 plays a protective role in cardiovascular diseases[13]. Therefore, drugs that target disease-specific regulatory mechanisms of HIF-1 activation are required.

Munc18-1-interacting protein 3 (Mint3) activates HIF-1 even under normoxic conditions by binding to FIH-1 and, thereby, interfering with the interaction between FIH-1 and HIF-1α[14,15]. Mint3 has a unique naturally disordered N-terminal region and a C-terminal region with one phosphotyrosine-binding and two PDZ domains conserved among other Mint family proteins, and the N-terminal region of Mint3 binds to FIH-1[14,16,17]. Mint3 requires the expression of a transmembrane metalloproteinase, MT1-MMP, to bind to FIH-1[15,18,19]; moreover, the mTORC1 signalling pathway and the monooxygenase NECAB3 promote the interaction between Mint3 and FIH-1 in cells from some types of cancer[20,21]. Mint3 is expressed ubiquitously, whereas MT1-MMP is highly expressed in cancer cells and in activated macrophages[15]. Thus, Mint3-mediated HIF-1 activation occurs only in these cells. Furthermore, the $K_m$ values of FIH-1 and PHDs for $O_2$ are about 90 and 230 μM, respectively, in vitro[22], indicating that Mint3-dependent HIF-1 activation can be observed under normoxic and moderate hypoxic conditions where FIH-1 can hydroxylate HIF-1α. Therefore, Mint3 is a unique molecule that activates HIF-1 in specific cell types within limited microenvironments. Reflecting these properties, Mint3 knockout (KO) mice show no apparent abnormality compared with HIF-1α knockout mice[23,24]. However, Mint3 KO mice showed resistance against acute inflammatory diseases, such as lipopolysaccharide (LPS)-induced endotoxic shock and influenza-induced acute pneumonia, and suppression of cancer metastasis due to attenuated hyperactivation of macrophage-lineage cells[23,25–27]. In cancer cells, Mint3 depletion suppresses tumour growth of various types of cancer, such as breast cancer, pancreatic cancer, lung cancer, and fibrosarcoma, and enhances chemosensitivity in cancer cells[18,20,21,28]. These results indicate that Mint3 inhibitors may present a new targeted therapeutic option for cancer and inflammatory diseases by avoiding severe adverse effects.

In this study, to develop Mint3 inhibitors, we first screened compounds that can bind to the N-terminal region of Mint3 using the chemical arrays of 23,275 compounds.

## Results

**Screening for small-molecule compounds that inhibit Mint3.** We initially surveyed compounds that bind to the N-terminus of Mint3 using a chemical array of 23,275 compounds (Fig. 1a) and identified one hit compound (Fig. 1b, compound #1). By using the HIF-1 reporter assay system (Fig. 1a) and HT1080 cells, where Mint3 activates HIF-1 even during normoxia, we evaluated whether compound #1 can suppress HIF-1 transcriptional activity[21]. Compound #1 showed an approximately 40% reduction in HIF-1 transcriptional activity at a concentration of 10 μM in HT1080 cells (Fig. 1c). To further explore potent Mint3 inhibitors, 18 compounds that were structurally similar to compound #1 were collected and subjected to the HIF-1 reporter assay in HT1080 cells (Supplementary Table 1). Compound #9 was excluded from further evaluation because it showed severe toxicity to HT1080 cells, and we observed, by using a microscope, that almost all the cells treated with compound #9 were detached from the plate. Compounds #5, #8, #11, and #19 significantly suppressed HIF-1 transcriptional activity (Fig. 1d). Among these compounds, compound #19, naphthofluorescein (Fig. 1e), showed the most profound suppression of HIF-1 transcriptional activity. Serial dilution of naphthofluorescein revealed that naphthofluorescein suppressed the HIF-1 reporter activity in a concentration-dependent manner (Fig. 1f). Therefore, we further assessed the biological actions of this compound.

**Naphthofluorescein disrupts Mint3–FIH-1 interaction in vitro and attenuates HIF-1 activity in cancer cells.** Direct interaction of Mint3 with FIH-1 activates HIF-1 function[14,18]. We used a pull-down assay to ascertain whether naphthofluorescein can block the interaction between Mint3 and FIH-1. Naphthofluorescein completely abolished the interaction between GST-FIH-1 and His$_6$-Mint3 in vitro (Fig. 2a). Next, the specificity of naphthofluorescein for Mint3 was examined by the HIF-1 reporter assay using control and Mint3 knockdown HT1080 cells (Fig. 2b). Naphthofluorescein reduced HIF-1 reporter activity by approximately 60% in control siRNA (siGFP)-transfected HT1080 cells (Fig. 2c, siGFP). In turn, Mint3 knockdown HT1080 cells showed reduced HIF-1 reporter activity, and naphthofluorescein did not further suppress the HIF-1 reporter activity in these cells (Fig. 2c, siMint3#1, #2). Overexpression of Mint3 and knockdown of FIH-1 also cancelled the suppression of HIF-1 reporter activity by naphthofluorescein (Fig. 2d–g). These results suggest that naphthofluorescein decreases HIF-1 reporter activity in a Mint3–FIH-1 axis-dependent manner.

Naphthofluorescein has been reported to inhibit the conversion of MT1-MMP from a pro-form to a mature form by furin, with an IC$_{50}$ of 12 μM [29]. Thus, we examined whether furin inhibition by naphthofluorescein affects HIF-1 activity in HT1080 cells. At a concentration of 10 μM, naphthofluorescein did not increase the pro-form of MT1-MMP in HT1080 cells, whereas furin inhibitors did not affect HIF-1 activity at a concentration that increased the pro-form of MT1-MMP in HT1080 cells (Furin-i I at 20 μM and Furin-i II at 10 μM) (Supplementary Fig. 1a, b). Mint3 has also been reported to control the cellular localisation of overexpressed furin[30]. However, neither Mint3 knockdown nor naphthofluorescein treatment affected endogenous furin localisation at the Golgi apparatus in HT1080 cells (Supplementary Fig. 1c, d). Thus, naphthofluorescein suppressed HIF-1 activity in HT1080 cells independent of furin inhibition under our experimental conditions. FIH-1 has been reported to hydroxylate another HIFα

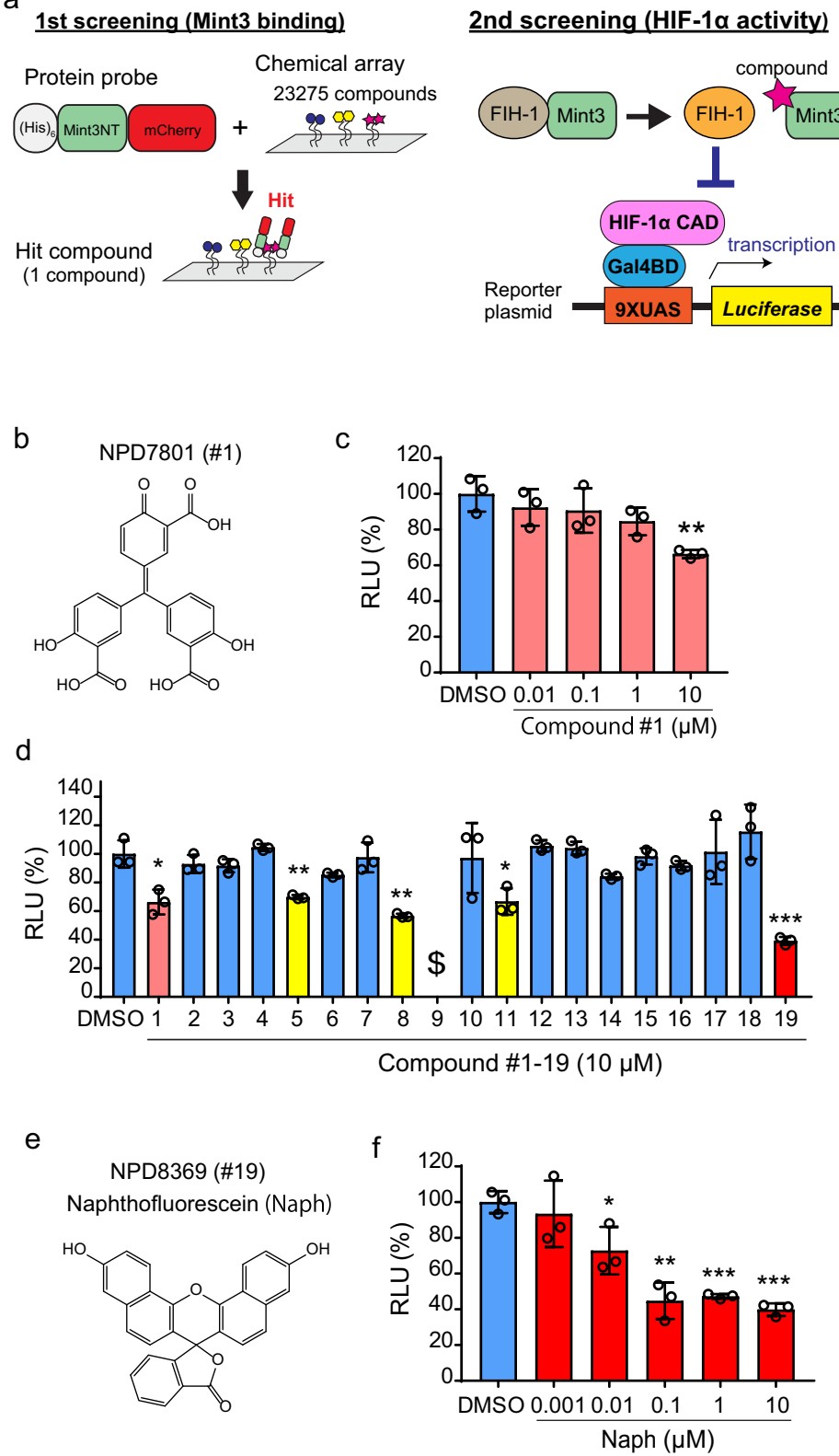

**Fig. 1 Screening for small-molecule compounds that inhibit Mint3. a** Schematic illustration of the screening process for Mint3 inhibitors. Mint3-binding compounds were identified using chemical arrays in the first screening. Then, using the luciferase assay in the second screening, hit compounds were tested to determine whether they inhibited HIF-1 transcriptional activity. **b** Structure of the hit compound, NPD7801 (compound #1). **c** Luciferase assay of HIF-1 activity in HT1080 cells treated with compound #1 at the indicated concentrations. **d** Luciferase assay of HIF-1 activity in HT1080 cells treated with 10 μM compound #1 and the analogues of compound #1 (#2–19). $ indicates no detection of luciferase activity due to cell death. **e** Structure of NPD8369 (compound #19), naphthofluorescein (Naph). **f** Luciferase assay for HIF-1 activity in HT1080 cells treated with DMSO or Naph at the indicated concentrations for 24 h. In (**c**), (**d**), and (**f**), error bars indicate SD ($n = 3$). Data were analysed using Student's $t$ test. *$p < 0.05$, **$p < 0.01$, ***$p < 0.001$.

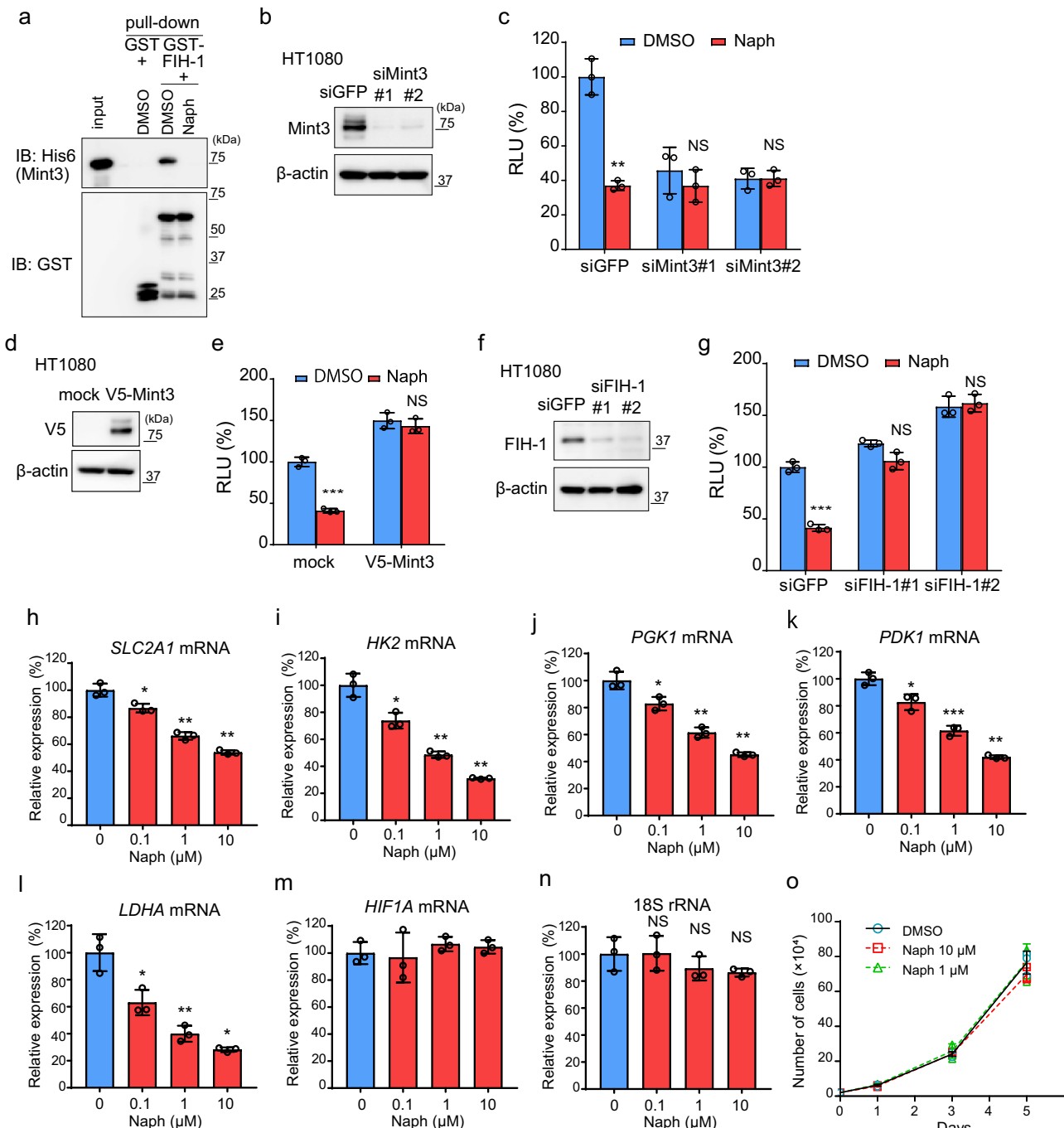

**Fig. 2 Naphthofluorescein disrupts Mint3–FIH-1 interaction and attenuates HIF-1 activity. a** Pull-down assay of GST-FIH-1 and His6-Mint3 in the presence of DMSO or naphthofluorescein (Naph; 10 μM). **b** Mint3 expression in HT1080 cells transfected with siRNAs against green fluorescent protein (GFP; siGFP) and Mint3 (siMint3#1, #2). **c** Luciferase assay for HIF-1 activity in siRNA-transfected HT1080 cells treated with DMSO or Naph (10 μM). **d** V5-tagged Mint3 expression in HT1080 cells transfected with mock or V5-tagged Mint3 expression vector. **e** Luciferase assay for HIF-1 activity in mock or V5-tagged-Mint3 expressing HT1080 cells treated with DMSO or Naph (10 μM). **f** Mint3 expression in HT1080 cells transfected with siRNAs against GFP (siGFP) and FIH-1 (siFIH-1#1, #2). **g** Luciferase assay for HIF-1 activity in siRNA-transfected HT1080 cells treated with DMSO or Naph (10 μM). **h–n** mRNA levels of glycolysis-related HIF-1 target genes (**h–l**) and *HIF1A* (**m**), and 18S rRNA levels (**n**) in HT1080 cells treated with DMSO or Naph at the indicated concentration for 24 h. **o** Cell growth assay of HT1080 cells treated with DMSO or Naph at the indicated concentrations. In (**c**), (**e**), (**g–o**), error bars indicate SD ($n = 3$). Data were analysed using Student's *t* test. $*p < 0.05$, $**p < 0.01$, $***p < 0.001$. NS not significant.

subunit, HIF-2α, much less effectively compared with HIF-1α due to differences in amino acid sequences around the target asparagine residue[22,31]. We also confirmed that overexpression of FIH-1 suppressed HIF-2α reporter activity less effectively than HIF-1α in HT1080 cells (Supplementary Fig. 1e, f). Consistent with these results, naphthofluorescein treatment also showed little

or no suppression of HIF-2α reporter activity in HT1080 cells (Supplementary Fig. 1g). Thus, naphthofluorescein shows selectivity for HIF-1 rather than for HIF-2.

Next, we examined whether naphthofluorescein affected the expression of HIF-1 target genes in HT1080 cells. Consistent with our observations in the HIF-1 reporter assay, naphthofluorescein

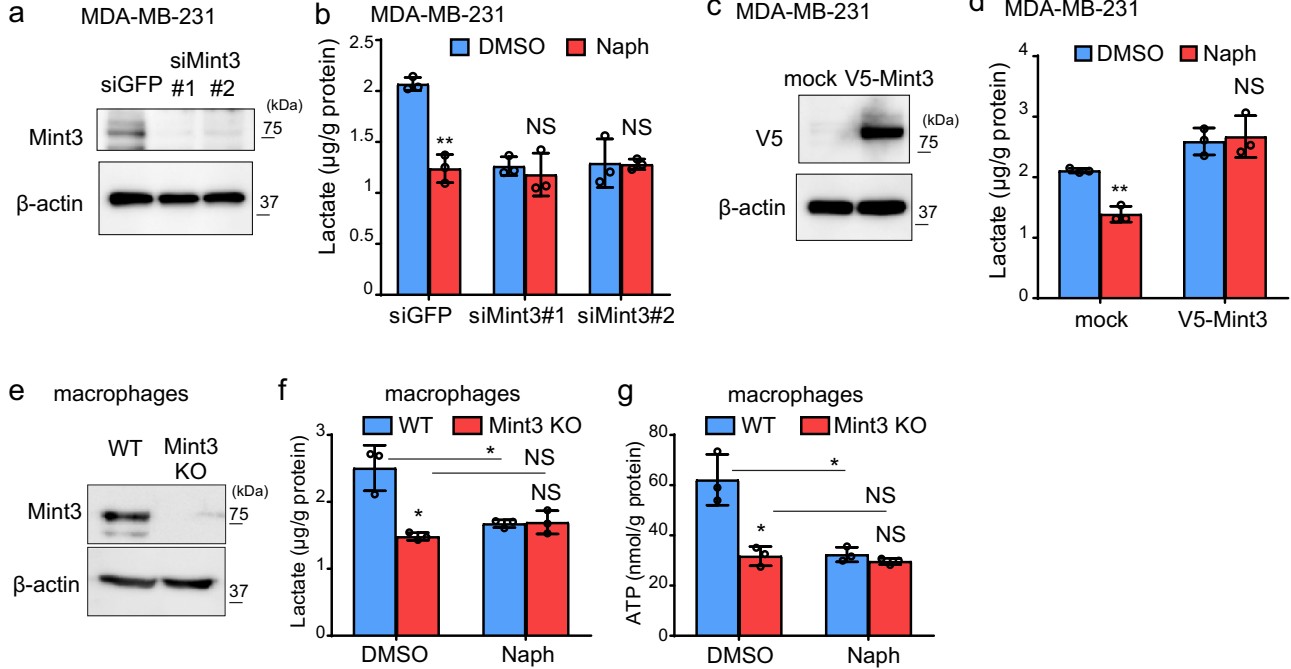

**Fig. 3 Naphthofluorescein attenuates Mint3-mediated glycolysis in MDA-MB-231 cells and macrophages. a** Mint3 expression in MDA-MB-231 cells transfected with siRNAs against green fluorescent protein (GFP; siGFP) and Mint3 (siMint3#1, #2). **b** Lactate production of control and Mint3 knockdown MDA-MB-231 cells treated with DMSO or naphthofluorescein (Naph; 10 μM). **c** V5-tagged Mint3 expression in MDA-MB-231 cells transfected with mock or V5-tagged Mint3 expression vector. **d** Lactate production of mock or V5-tagged Mint3 expressing MDA-MB-231 cells treated with DMSO or Naph (10 μM). **e** Mint3 expression in wild-type (WT) and Mint3 knockout (KO) bone-marrow-derived macrophages. **f** Lactate production of WT and Mint3 KO macrophages treated with DMSO or Naph (10 μM). **g** ATP levels of WT and Mint3 KO macrophages treated with DMSO or Naph (10 μM). In (**b**), (**d**), (**f**), and (**g**), error bars indicate SD ($n = 3$). Data were analysed using Student's t test. *$p < 0.05$, **$p < 0.01$. NS not significant.

suppressed the expression of HIF-1-target glycolysis-related genes in HT1080 cells in a concentration-dependent manner, whereas HIF-1α mRNA and 18S rRNA expressions were unaffected (Fig. 2h–n). FIH-1 knockdown also attenuated the suppression of HIF-1 target gene expression by naphthofluorescein in HT1080 cells (Supplementary Fig. 1h–m), indicating that naphthofluorescein suppressed HIF-1 target gene expression via the Mint3–FIH-1 axis. Mint3 depletion attenuates the in vivo growth of HT1080 cells but does not affect cell proliferation in vitro[20,21]. Likewise, at a concentration of 10 μM, naphthofluorescein did not affect the proliferation of HT1080 cells but suppressed HIF-1 reporter activity and the expression of endogenous glycolysis-related genes (Fig. 2o). Taken together, naphthofluorescein inhibits the Mint3-dependent transcriptional activity of HIF-1 without causing non-specific cell toxicity in HT1080 cells.

**Naphthofluorescein attenuates Mint3-mediated glycolysis in MDA-MB-231 cells and macrophages**. In addition to HT1080 cells, Mint3 promotes glycolysis in breast cancer MDA-MB-231 cells and in murine macrophages[18,23,32]. Thus, by measuring the levels of lactate, an end product of glycolysis, we examined whether naphthofluorescein can suppress glycolysis in these cells. Mint3 knockdown (siMint3#1, #2) MDA-MB-231 cells showed reduced lactate production compared to control (siGFP) cells (Fig. 3a, b), as previously reported[18]. Naphthofluorescein decreased lactate production in control MDA-MB-231 cells to the levels in Mint3 knockdown cells but did not further affect the lactate levels of Mint3 knockdown cells (Fig. 3b). Overexpression of exogenous Mint3 cancelled the decreased lactate production by naphthofluorescein in MDA-MB-231 cells (Fig. 3c, d). Similar to MDA-MB-231 cells, that naphthofluorescein decreased lactate production in WT macrophages to the levels in Mint3 KO cells,

but did not further affect the lactate levels of Mint3 KO cells (Fig. 3e, f), show the reduced ATP levels due to defects of glycolysis[23,26,32]. Furthermore, naphthofluorescein reduced ATP levels in WT macrophages but not in Mint3 KO cells (Fig. 3g). Thus, naphthofluorescein suppresses Mint3-dependent glycolysis in MDA-MB-231 cells and in macrophages.

**Administration of naphthofluorescein attenuates the interaction between Mint3 and FIH-1 and the expression of HIF-1 target genes in vivo**. To understand the in vivo effects of naphthofluorescein, we first evaluated the acute toxicity of naphthofluorescein in mice. The experiments were conducted in accordance with the institutional ethical guidelines for animal experiments. Naphthofluorescein 100 mg/kg body weight (b.w.) was intraperitoneally injected into C57BL/6 mice for 2 weeks (5-day/2-day injection-rest cycle per week). The mice receiving naphthofluorescein did not show weight loss (Fig. 4a) or apparent histological abnormalities in the lung, liver, and kidney (Supplementary Fig. 2a). Blood biochemical analyses did not show any significant difference between vehicle- and naphthofluorescein-treated mice (Supplementary Table 2). Thus, the administration of naphthofluorescein does not cause severe adverse effects for at least 2 weeks in mice.

Next, we addressed the question of whether naphthofluorescein can block the interaction between Mint3 and FIH-1 even in vivo. Nude mice with tumours of HT1080 cells expressing FLAG-tagged FIH-1 were injected with vehicle or naphthofluorescein, FLAG-tagged FIH-1 was immunoprecipitated from tumour lysates 3 h after the injection, and co-precipitated Mint3 was detected by western blotting. Administration of naphthofluorescein indeed attenuated the interaction between FLAG-tagged FIH-1 and Mint3 even in vivo (Fig. 4b). We also confirmed that

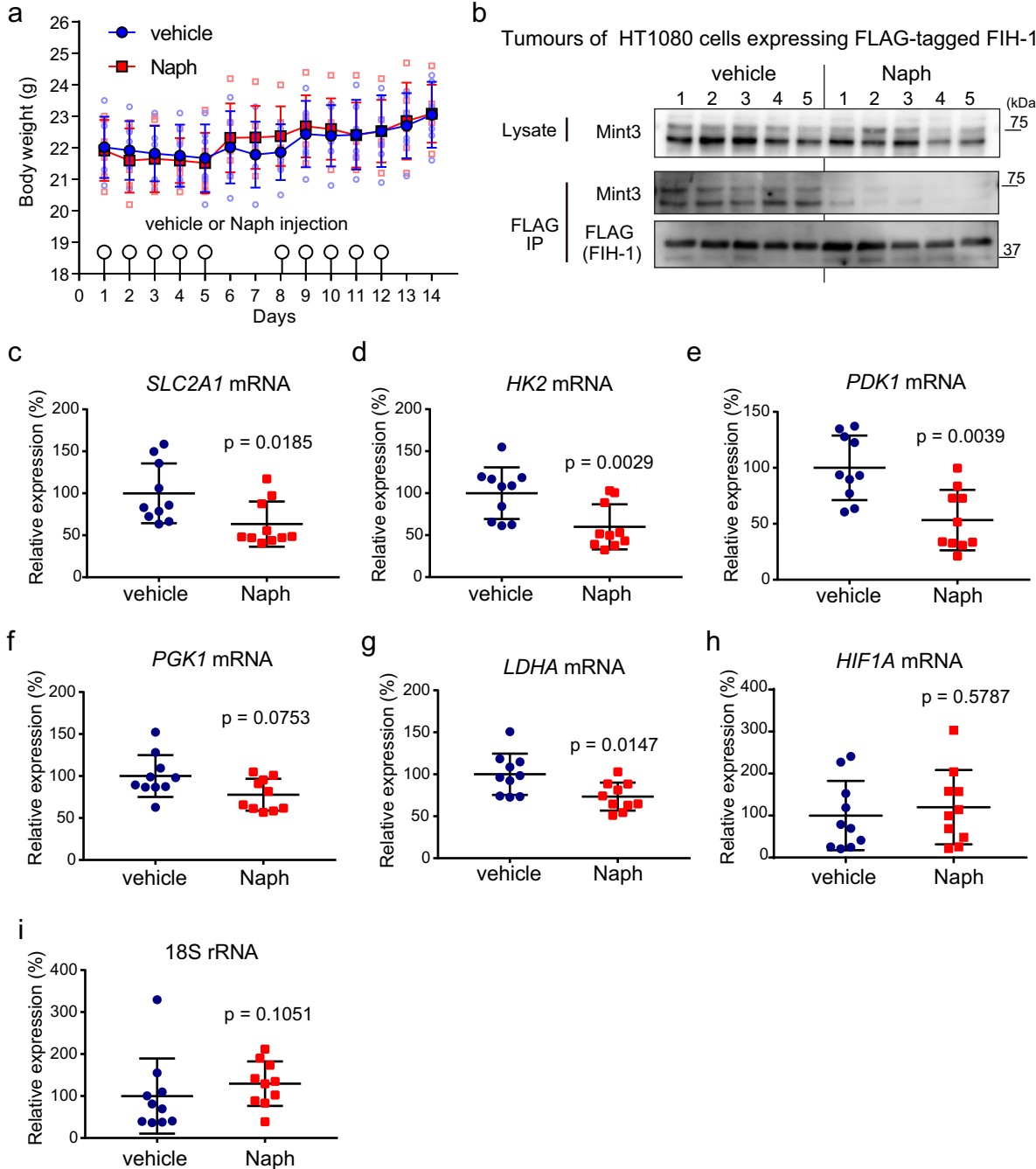

**Fig. 4 Administration of naphthofluorescein attenuates the interaction between Mint3 and FIH-1 and the expression of HIF-1 target genes in tumour xenografts. a** Body weight of mice with vehicle or naphthofluorescein (Naph; 100 mg/kg b.w.) injection ($n = 6$ per group). **b** Administration of Naph attenuates the interaction between Mint3 and FIH-1 in tumours. Tumours of HT1080 cells expressing FLAG-tagged FIH-1 were collected from mice injected vehicle or Naph ($n = 5$ per group) 3 h after the injection. FLAG-tagged FIH-1 was immunoprecipitated (IP FLAG) from tumour lysates, and precipitates were analysed by western blotting using the indicated antibodies. **c–i** Expression of HIF-1 target genes (**c–g**), *HIF1A* (**h**), and 18S rRNA (**i**) in tumours of HT1080 cells 6 h after Naph injection. Error bars indicate the SD ($n = 10$). Data were analysed using the Mann–Whitney U test.

the administration of naphthofluorescein did not affect the conversion of the potential furin substrates MT1-MMP and integrin α5 [33,34] from their pro-forms to the mature forms in these tumours (Supplementary Fig. 2b).

Subsequently, we examined whether naphthofluorescein can suppress the expression of HIF-1 target genes in tumours. Nude mice with tumours of HT1080 cells were injected with vehicle or naphthofluorescein, and gene expression in tumours was analysed 8 h after the injection. The expression of the tested HIF-1 target genes was apart from PGK1 significantly reduced in tumours from naphthofluorescein-injected mice compared to that in tumours from vehicle-injected mice, and HIF-1α mRNA levels were not affected in these tumours (Fig. 4c–i). In turn, naphthofluorescein administration could not decrease the expression of these genes in tumours of Mint3-overexpressing HT1080 cells (Supplementary Fig. 2c–g). These results indicate that naphthofluorescein suppresses the expression of HIF-1-target glycolysis-related genes by targeting Mint3 in vivo.

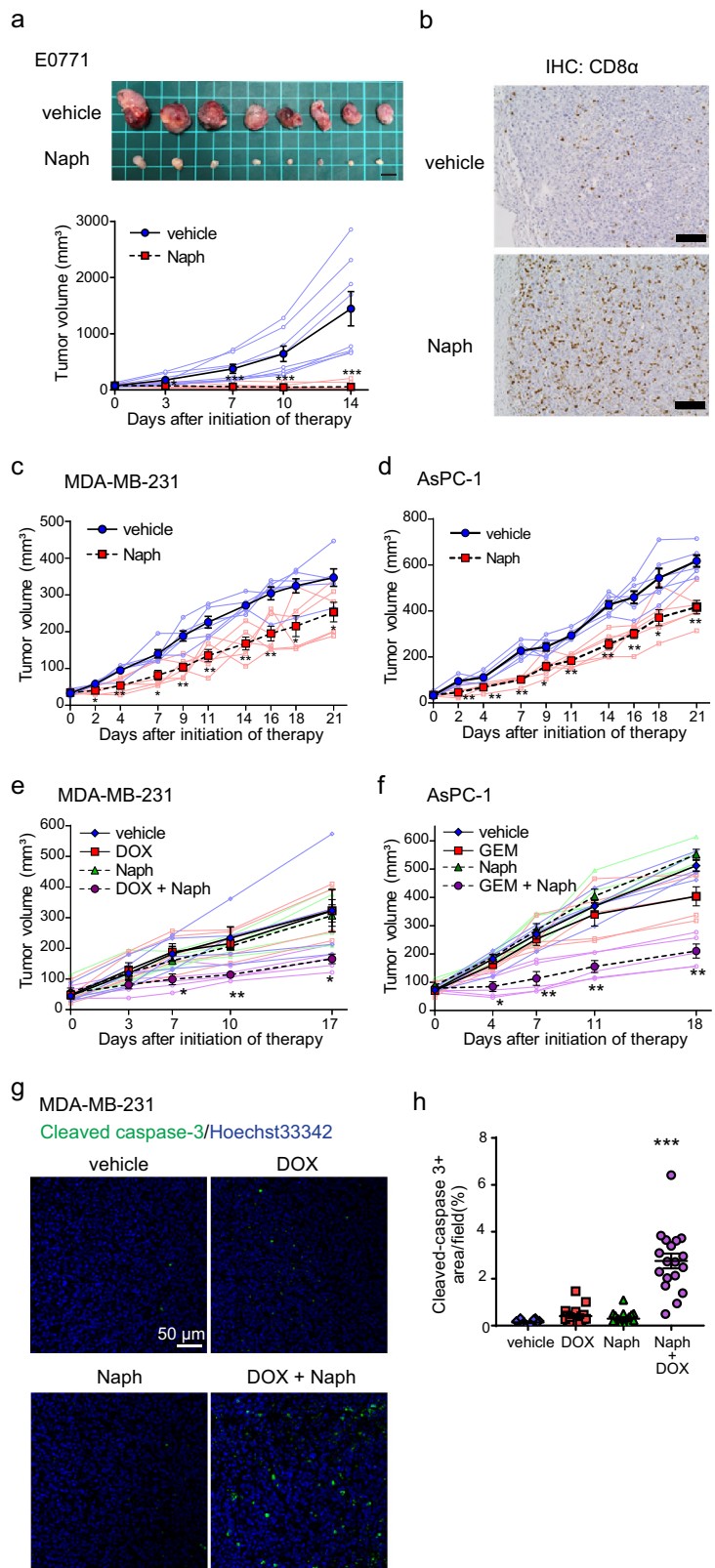

**Naphthofluorescein attenuates tumour growth and augments the effect of anticancer drugs in vivo.** Mint3 knockdown attenuates the tumour growth of various types of cancers, such as breast cancer, pancreatic cancer, lung cancer, and fibrosarcoma, in mice[18,20,21,28]. Thus, we first examined whether naphthofluorescein suppresses the tumour growth of murine triple-negative breast cancer E0771 cells in immunocompetent C57BL/

6 mice. Naphthofluorescein administration for 5 days a week strikingly suppressed the tumour growth of subcutaneously injected E0771 cells (Fig. 5a). By contrast, in immunodeficient nude mice, suppression of E0771 tumour growth by naphthofluorescein was moderate compared to that observed in immunocompetent mice (Supplementary Fig. 3), implying T-cell-dependent and -independent anti-tumour effects of naphthofluorescein. Further

**Fig. 5 Naphthofluorescein attenuates tumour growth and boosts the effects of anticancer drugs in vivo. a** Tumour growth of E0771 cells in C57BL/6 mice ($n = 8$) treated with vehicle or naphthofluorescein (Naph; 100 mg/kg b.w.). Mice were intraperitoneally injected with vehicle or Naph five times a week (top). Photos of tumours at day 14 after initiation of therapy. Scale bar = 1 cm (bottom); tumour volumes of E0771 cells in mice. **b** Immunohistochemical (IHC) analysis of CD8α in tumours of E0771 cells treated with vehicle or Naph. Scale bars = 100 μm. **c, d** Tumour growth of MDA-MB-231 (**c**) and AsPC-1 (**d**) cells in immunodeficient mice treated with vehicle or Naph (100 mg/kg b.w.). Mice were intraperitoneally injected with vehicle or Naph five times a week ($n = 6$). **e, f** Tumour growth of MDA-MB-231 (**e**) and AsPC-1 (**f**) cells in immunodeficient mice treated with vehicle or Naph (100 mg/kg b.w.) in combination with vehicle, doxorubicin (DOX; 2 mg/kg b.w.), or gemcitabine (GEM; 20 mg/kg b.w.) twice a week ($n = 5$). **g, h** Immunostaining of cleaved caspase-3 in tumours of MDA-MB-231 from vehicle- or Naph-treated mice in combination with DOX at day 14 after initiation of therapy. **g** Representative photos. **h** Cleaved caspase-3-positive areas were counted in tumour sections ($n = 18$, from six tumours per group). In (**a**), (**c–f**), and (**h**), error bars indicate the SEM. Data were analysed using the Mann–Whitney $U$ test. *$p < 0.05$, **$p < 0.01$, ***$p < 0.001$.

assessment of the T-cell-dependent anti-tumour effects of naphthofluorescein showed that more CD8$^+$ T cells infiltrated the tumours of naphthofluorescein-administered immunocompetent mice compared with tumours of vehicle-administered mice (Fig. 5b).

Next, naphthofluorescein was tested in xenograft models of human triple-negative breast cancer MDA-MB-231 and pancreatic cancer AsPC-1 cells. The administration of naphthofluorescein for 5 consecutive days per week significantly attenuated tumour growth of MDA-MB-231 and AsPC-1 cells in immunodeficient mice (Fig. 5c, d). In turn, naphthofluorescein did not attenuate tumour growth of FIH-1-depleted MDA-MB-231 cells (Supplementary Fig. 3b–d). Thus, naphthofluorescein can suppress tumour growth in human cancer cells in an FIH-1-dependent manner. HIF-1 is involved in chemoresistance, and combinations of HIF-1 inhibitors and chemotherapy show strong anti-tumour effects[4,6]. In addition, Mint3 depletion suppresses chemoresistance in pancreatic cancer cells[28]. Therefore, we examined whether the combination of naphthofluorescein and chemotherapy showed additive/synergistic effects on tumour suppression. Naphthofluorescein was injected into tumour-bearing mice with or without cytotoxic chemotherapeutic reagents (doxorubicin for MDA-MB-231 and gemcitabine for AsPC-1 cells) twice a week. In these treatment regimens, naphthofluorescein or chemotherapeutic agents alone did not effectively suppress tumour growth. However, the combination of naphthofluorescein with doxorubicin or gemcitabine strikingly suppressed the tumour growth of MDA-MB-231 and AsPC-1 cells, respectively (Fig. 5e, f). We confirmed that the combined administration of naphthofluorescein and doxorubicin remarkably increased the apoptotic marker cleaved caspase-3-positive area in tumours of MDA-MB-231 cells (Fig. 5g, h). Taken together, naphthofluorescein improves the efficacy of chemotherapy in xenograft models.

**Naphthofluorescein attenuates host Mint3-mediated lung metastasis.** Metastasis involves not only cancer cells but also host cells such as immune and endothelial cells[35–38]. Mint3 in host cells, especially in inflammatory monocytes, promotes metastatic niche formation in the lung[25,26]. Thus, using experimental lung metastasis models of mouse melanoma B16F10 cells, we examined whether naphthofluorescein inhibits host Mint3-dependent metastasis (Fig. 6a). Naphthofluorescein significantly suppressed lung metastasis of B16F10 cells compared to vehicle (Fig. 6b, c) in a dose-dependent manner (Fig. 6d). Similarly, naphthofluorescein suppressed lung metastasis of mouse lung cancer Lewis lung carcinoma (LLC) cells (Supplementary Fig. 4a, b). Importantly, naphthofluorescein did not further suppress the lung metastasis of B16F10 cells in Mint3 KO mice (Fig. 6e, f), indicating that the effect of naphthofluorescein on metastasis is attributable to host Mint3, which promotes metastatic niche formation in the lung by recruiting inflammatory monocytes and inducing the expression of E-selectin in endothelial cells that are adjacent to cancer cells in metastasis-affected lungs[26]. Thus, we examined whether naphthofluorescein suppresses this metastatic

niche formation. Naphthofluorescein suppressed both recruitment of inflammatory monocytes and E-selectin expression in metastatic lung cancer (Fig. 6g, h). Mint3 depletion attenuates chemotaxis toward CCL2 in macrophages/inflammatory monocytes due to the defect in ATP production via glycolysis without affecting the expression levels of CCR2 in macrophages, those of CCL2 in metastatic lungs, and serum in tumour-inoculated mice[23,26]. Similarly, naphthofluorescein did not affect the expression levels of CCR2 in macrophages, those of CCL2 in metastatic lungs, and serum of tumour-inoculated mice (Supplementary Fig. 4c–e). Thus, naphthofluorescein administration phenocopies the host Mint3 depletion in lung metastatic niche formation.

**Naphthofluorescein attenuates endotoxic shock and inflammatory cytokine production in macrophages.** Naphthofluorescein inhibited host Mint3 in experimental metastasis models. This prompted us to evaluate the effects of naphthofluorescein in other Mint3-related disease models. Mint3 KO mice are resistant to LPS-induced endotoxic shock due to decreased macrophage hyperactivation[23]. Thus, we examined whether the administration of naphthofluorescein attenuates LPS-induced endotoxic shock. Pretreatment with naphthofluorescein significantly improved survival of LPS-injected WT mice, similar to that of mice with genetic depletion of Mint3 (Fig. 7a, b). Furthermore, naphthofluorescein pretreatment decreased serum levels of inflammatory cytokines, such as tumour necrosis factor-alpha (TNF-α), interleukin (IL)-6, and IL-12p70, after LPS injection in WT mice (Fig. 7c–e), similar to that in the Mint3 KO mice phenotype[23]. Supporting these in vivo observations, naphthofluorescein decreased TNF-α and IL-6 production in LPS-stimulated WT macrophages to levels observed in Mint3 KO macrophages but did not further decrease cytokine production in LPS-stimulated Mint3 KO macrophages (Fig. 7f, g). Thus, naphthofluorescein specifically inhibits Mint3-dependent inflammatory cytokine production in macrophages, and naphthofluorescein pretreatment attenuates LPS-induced cytokine storm and improves the survival of mice with LPS-induced endotoxic shock.

## Discussion

We surveyed the range of small-molecule compounds that inhibit Mint3-mediated HIF-1 activation and identified naphthofluorescein, which blocked the interaction between Mint3 and FIH-1 in vitro and suppressed Mint3-dependent HIF-1 activity and glycolysis in cancer cells and macrophages without causing cytotoxicity in vitro. In vivo, naphthofluorescein decreased the expression of HIF-1 target genes in tumours without apparent adverse effects. Furthermore, naphthofluorescein administration suppressed tumour growth, metastasis, and LPS-induced endotoxic shock in mice. Especially, naphthofluorescein did not show any effects in Mint3 KO mice on metastasis and endotoxic shock models and on HIF-1 target gene expression in tumours of

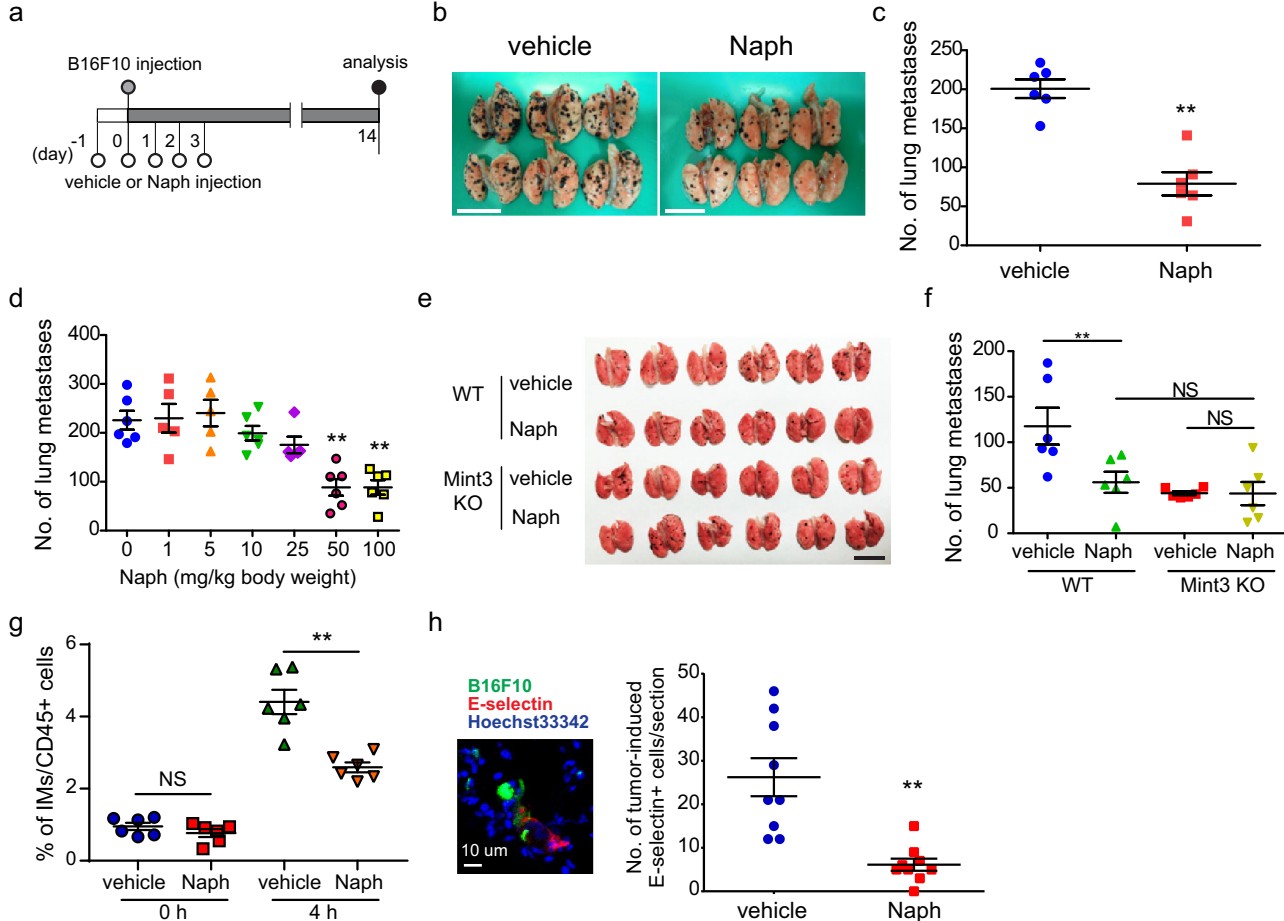

**Fig. 6 Naphthofluorescein attenuates host Mint3-mediated lung metastasis. a** Schematic illustration of the experimental lung metastasis assay of B16F10 cells. **b** Photos of metastatic lungs from vehicle- or naphthofluorescein-treated (Naph; 100 mg/kg b.w.) mice. Scale bars = 1 cm. **c** Number of metastatic foci of B16F10 cells in the lungs from mice treated with vehicle or Naph at the indicated amount (n = 6 mice per group). **d** Number of metastatic foci of B16F10 cells in the lungs from mice treated with Naph at the indicated dose (n = 5–6 mice per group). **e** Photos of metastatic lungs from vehicle- or Naph-treated WT and Mint3 KO mice. Scale bar = 1 cm. **f** Number of metastatic foci of B16F10 cells in the lungs from vehicle- or Naph-treated (100 mg/kg b.w.) WT and Mint3 KO mice (n = 6 mice per group). **g** Flow cytometric analysis of inflammatory monocytes (IMs) in the lungs at 4 h after intravenous injection of B16F10 cells into vehicle- or Naph-treated (100 mg/kg b.w.) mice (n = 6 per group). **h** Immunostaining analysis of E-selectin (red) expression in the lungs of vehicle- or Naph-treated mice 6 h after intravenous injection with fluorescein-labelled B16F10 cells (green). Nuclei are stained with Hoechst 33342 (blue). (Left) Representative photos of lungs from vehicle-treated mice. (Right) Quantitative analysis of E-selectin-positive cells in the lungs (n = 9, from three tumours per group). In (**c**), (**d**), and (**f**–**h**), data are presented as the mean ± SEM. Data were analysed using the Mann–Whitney U test. **p < 0.01. NS not significant.

Mint3-overexpressing HT1080 cells, indicating the specificity of naphthofluorescein for Mint3 in vivo (Fig. 7h). Mint3 enhances the transcriptional activity of HIF-1α by suppressing the HIF-1 inhibitor FIH-1 [14,15]. Although their pathophysiological roles are not well understood, FIH-1 can hydroxylate not only HIF-1α but also other proteins with ankyrin repeat domains, such as Notch and IκBα[39–42]. Thus, Mint3 might suppress the hydroxylation of these abovementioned proteins by FIH-1; however, we cannot exclude the possibility that FIH-1 target proteins other than HIF-1α partially contribute to the anti-tumour and anti-inflammatory effects of naphthofluorescein. Naphthofluorescein suppressed tumour growth of MDA-MB-231 in an FIH-1 dependent manner. In turn, naphthofluorescein suppressed host Mint3-mediated metastasis and LPS-induced endotoxic shock in vivo, but we could not completely exclude in this study the possibility that naphthofluorescein might suppress in vivo Mint3-mediated metastasis and endotoxic shock independently from FIH-1. Further studies using FIH-1 knockout or conditional knockout mice may clarify the specificity of naphthofluorescein effects on

FIH-1 in metastasis and endotoxic shock models. Naphthofluorescein and Mint3 have been reported to control furin activity/localisation, though we could not reproduce these effects in our experimental conditions[29,30]. Furin is involved in various biological events by targeting various substrates including MT1-MMP and integrin α5, and furin knockout in mice is embryonically lethal[33,34,43]. In turn, Mint3 knockout mice show no apparent abnormality[23,24,44]. Thus, Mint3 is not likely an essential regulator of furin, at least not during embryogenesis. However, we do not exclude the possibility that furin inhibition partially contributed to the anti-tumour effects of naphthofluorescein in Mint3-dependent and -independent manners (Fig. 7h).

Intriguingly, naphthofluorescein suppressed the tumour growth of E0771 cells in an immunity-dependent manner (Fig. 5a, b). The mechanism whereby Mint3 contributes to anti-tumour immunity remains unclear; however, Mint3 controls the inflammatory function of monocytes and, thereby, promotes metastasis[25,26]. Thus, Mint3 in macrophage-lineage cells might

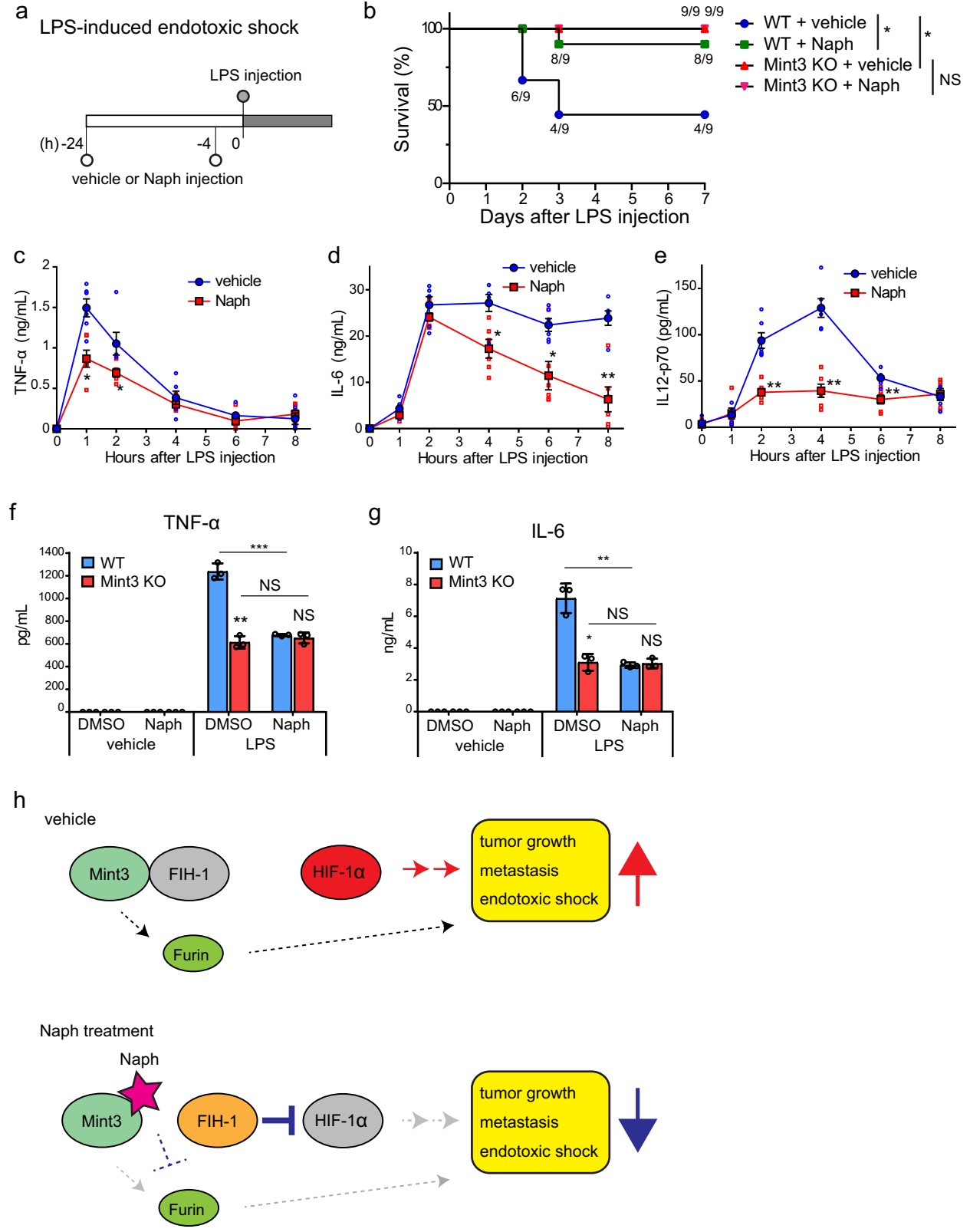

contribute to anti-tumour immunity. In tumour tissues, tumour-associated macrophages (TAMs) polarise to the M2-like subtype which suppresses anti-tumour immunity[45]. HIF-1α promotes M2-like polarisation in TAMs[46], implying that Mint3 might also control M2-like polarisation in TAMs through HIF-1 activation, which we are investigating currently. In contrast, Mint3 depletion in cancer cells attenuates tumour growth[18,20,21,28]. Thus, the anti-

tumour effects of naphthofluorescein that were observed in immunodeficient mice seem to mainly reflect the effects of Mint3 inhibition in cancer cells. However, the suppression of tumour growth of MDA-MB-231 cells by naphthofluorescein was moderate compared with the results in a previous report on Mint3 depletion in MDA-MB-231 cells[18], which implies insufficient concentration and persistence duration of naphthofluorescein

**Fig. 7 Naphthofluorescein attenuates LPS-induced endotoxic shock. a** Schematic illustration of LPS-induced endotoxic shock experiments. **b** Survival of LPS-injected mice treated with vehicle or naphthofluorescein (Naph; 100 mg/kg b.w.; $n = 9$ per group). Data were analysed using the log-rank test. $*p < 0.05$. NS not significant. **c–e** Serum TNF-α (**c**), IL-6 (**d**), and IL-12p70 (**e**) levels from LPS-injected mice treated with vehicle or Naph. Data are presented as the mean ± SEM ($n = 6$). Data were analysed using the Mann–Whitney $U$ test. $*p < 0.05$, $**p < 0.01$. **f, g** TNF-α (**f**) and IL-6 (**g**) production in LPS-stimulated WT and Mint3 KO macrophages with or without Naph (10 μM). Data are presented as the mean ± SD ($n = 3$). Data were analysed using Student's $t$ test. $*p < 0.05$, $**p < 0.01$, $***p < 0.001$. NS not significant. **h** Schematic diagram illustrating how Naph attenuates tumour growth, metastasis, and endotoxic shock. Naph disrupts the interaction between Mint3 and FIH-1, and liberated FIH-1 inhibits HIF-1 activity in cancer cells and macrophages, resulting in attenuation of tumour growth, metastasis, and endotoxic shock in vivo. Direct and indirect furin suppression by Naph might also contribute to these in vivo effects.

within tumour tissues. In addition, naphthofluorescein is less water-soluble and unsuitable for oral administration. Further modifications of naphthofluorescein and/or combination with appropriate drug delivery systems might improve the efficacy of anti-tumour effects and provide other options for routes of administration, including oral administration.

Many studies have demonstrated that HIF-1 inhibition is an effective strategy for cancer therapy in various experimental models, and our results of an inhibition of Mint3-mediated HIF-1 activation via FIH-1 by naphthofluorescein parallel these reports[6]. In turn, PHD inhibitors have been reported to show anti-tumour effects in some experimental models[47–49]. PHD inhibitors induce the accumulation of not only HIF-1α but also HIF-2α and increase erythropoiesis via HIF-2 accumulation in the kidney[50]. This might improve hypoxic conditions in tumours, resulting partly in the anti-tumour effects. In addition, HIF activities must be controlled properly for tumour progression in response to the tumour microenvironment. Thus, aberrant accumulation of HIFs caused by PHD inhibitors might also show anti-tumour effects similar to those of HIF inhibition. Clinical trials on the HIF-2-specific inhibitor PT2385 and its derivative PT2977 are ongoing, whereas highly specific HIF-1 inhibitors are currently being developed[51,52]. HIF-1 is essential not only for cancer and inflammatory diseases but also for the homoeostasis of normal organs; HIF-1α KO mice show diverse abnormalities in various organs[7–12]. In addition, the protective roles of HIF-1 in cardiovascular diseases imply the contraindication of systemic HIF-1 inhibition in individuals with heart failure[13]. Thus, it might be difficult for HIF-1-specific inhibitors to avoid adverse effects. In contrast, Mint3 activates the transcriptional activity of HIF-1α in specific types of cells, such as cancer cells and macrophages, by suppressing FIH-1, and Mint3 KO mice showed no apparent abnormality[23,24], which promises less adverse effects with Mint3 inhibitors. Naphthofluorescein suppressed tumour growth in an immunity-dependent manner and augmented the efficacy of cytotoxic anticancer drugs. Thus, Mint3 inhibitors are expected to be used not only individually, but also in combination with currently approved immunotherapy or cytotoxic reagents. In addition to its effects in cancer, naphthofluorescein showed anti-inflammatory effects in LPS-induced endotoxic shock. With regard to its suppression of macrophage hyperactivation, it is interesting to understand whether Mint3 inhibitors are available for modulating the cytokine storm in the coronavirus disease (COVID-19)[53,54] and other chronic inflammatory diseases, such as rheumatoid arthritis and type-2 diabetes. We identified one compound that bound to Mint3 and weakly inhibited HIF-1 activity in cancer cells. Analogues of this compound were then evaluated and, among them, naphthofluorescein showed the highest HIF-1 suppression activity. Further analyses revealed that naphthofluorescein inhibited Mint3–FIH-1 interaction in vitro, the expression of HIF-1 target genes in cancer cells, and Mint3-dependent glycolysis in cancer cells and macrophages. The administration of naphthofluorescein suppressed tumour growth, metastasis, and endotoxic shock in vivo without apparent adverse

effects. Overall, our work characterised Mint3 as a potential therapeutic target in cancer and inflammatory diseases, and provides insights into the effectiveness of novel approaches that combine Mint3 inhibitors and other therapeutic modalities.

In conclusion, we identified naphthofluorescein, which interferes with the protein–protein interaction between Mint3 and FIH-1 in vitro and suppresses tumour growth, metastasis, and LPS-induced endotoxic shock. Derivatives of naphthofluorescein and other compounds that target Mint3 might be useful therapeutic candidates for cancer and inflammatory diseases.

## Methods

**Compounds.** For in vitro experiments, naphthofluorescein and other compounds for the chemical array screening were provided by RIKEN NPDepo (Saitama, Japan). For in vivo experiments, naphthofluorescein was synthesised by the Sundia MediTech Company (Shanghai, China) at a purity > 95%. Furin inhibitors I and II were purchased from Merck (Kenilworth, NJ, USA).

**Chemical array screening.** The chemical arrays were prepared by a previously described method[55]. We used 23,275 compounds from RIKEN NPDepo for the chemical arrays. Chemical array screening was conducted as previously described[56,57]. We used a 4-μM concentration of His$_6$-Mint3-mCherry in the chemical arrays.

**Cell culture.** B16F10 melanoma and LLC cells were obtained from the Cell Resource Center for Biomedical Research, Institute of Development, Aging and Cancer, Tohoku University (Miyagi, Japan). HT1080, MDA-MB-231, and AsPC-1 cells were obtained from the American Type Culture Collection (Manassas, VA, USA). HT1080 cells expressing FLAG-tagged FIH-1 or V5-tagged Mint3 were established using pLenti6 lentiviral vectors (Thermo Fisher Scientific) as previously described[28,58]. Cells were cultured in Dulbecco's modified Eagle medium (DMEM; HT1080, MDA-MB-231, B16F10, and LLC) or RPMI1640 medium (AsPC-1; both from Thermo Fisher Scientific) containing 10% fetal bovine serum (FBS), 100 units/mL penicillin, and 100 μg/mL streptomycin at 37 °C in a humidified incubator with 5% $CO_2$. shLacZ- and shFIH-1-expressing MDA-MB-231 cells were prepared as previously described[18]. The sequences of shRNAs are listed in Table 1. Bone-marrow-derived macrophages were obtained by previously described methods[27]. All cell lines were routinely tested to exclude mycoplasma contamination.

**siRNA knockdown.** siRNA knockdown was carried out using Lipofectamine RNAiMAX (Thermo Fisher Scientific) as described earlier[59]. The siRNA sequences for each gene are described in Table 1.

**Reporter assay.** The reporter assay was performed as previously described, but with minor modifications[14,21]. A pGL4.35 reporter vector containing the firefly *luciferase* gene under the control of a transcriptional regulatory unit comprising 9× Gal4-binding elements was purchased from Promega. A pRL vector expressing *Renilla* luciferase (Promega) served as an internal control. The HT1080 cells ($1 \times 10^6$) were seeded in a 90-mm dish and co-transfected with reporter plasmid (2.5 μg), internal control vector (250 ng), and pcDNA3 Gal4BD-HIF-1α CAD (727–826 amino acids) or pcDNA3 Gal4BD-HIF-2α CAD (769–870 amino acids) plasmid[14,21] (250 ng) using Lipofectamine 2000 (Thermo Fisher Scientific). Twenty-four hours after transfection, the cells ($1 \times 10^4$/well) were seeded in 96-well plates and treated with compounds at the indicated concentrations for 24 h. For the overexpression experiments, 50 ng/well of pcDNA3 V5-tagged Mint3, pcDNA3 V5-tagged FIH-1, or mock plasmid were transfected to HT1080 cells 2 h before compound treatment. Luciferase activity was measured using the Dual-Glo Luciferase Assay System (Promega) according to the manufacturer's instructions. Luminescence was measured using a GloMax 20/20 luminometer (Promega).

| Table 1 shRNA and siRNA sequences. | |
|---|---|
| | Sequence |
| shLacZ | 5′- GCTACACAAATCAGCGATTTCGAAAAATCGCTGATTTGTGTAG-3′ |
| shFIH-1#1 | 5′- GGAAGATTGTCATGGACTTCTCGAAAGAAGTCCATGACAATCTTCC-3′ |
| shFIH-1#2 | 5′- GGATTACCATCACTGTGAACTCGAAAGTTCACAGTGATGGTAATCC-3′ |
| siGFP | 5′-AUCCGCGCGAUAGUACGUA -3′ |
| siMint3#1 | 5′-CCUAUGGCGAGGUGCAUAU-3′ |
| siMint3#2 | 5′-GGUUCUUGGUCCUGUAUGA-3′ |
| siFIH-1#1 | 5′-GCCAAUUUCCAGAACUUUA-3′ |
| siFIH-1#2 | 5′-GUAUUGCACGCUGCACUUA-3′ |

**Preparation of recombinant proteins and GST pull-down assay.** Recombinant proteins such as GST, GST-FIH-1, His$_6$-Mint3NTmCherry, and His$_6$-Mint3 were prepared as previously described[14,18], and GST pull-down assays were performed as previously described, but with minor modifications[14,18]. Briefly, glutathione–Sepharose 4B (GE Healthcare)-conjugated GST fusion proteins (~1 µg) were preincubated with 0.5 mg/mL bovine serum albumin (BSA) in lysis buffer (150 mM NaCl, 50 mM Tris pH 8.0, 1% NP-40) for 30 min at 4 °C. Thereafter, 1 µg His$_6$-tagged FIH-1 was resuspended in lysis buffer containing 0.5 mg/mL BSA and then added to the beads. After mixing on a rotator for 2 h, the beads were washed four times with lysis buffer. The proteins were eluted with Laemmli sample buffer and analysed by sodium dodecyl sulfate-polyacrylamide gel electrophoresis (SDS-PAGE), followed by western blotting analysis.

**Western blotting analysis.** Cells were lysed with lysis buffer and centrifuged at $20,000 \times g$ for 15 min at 4 °C. The supernatants were collected, and the total protein content was measured using a Bradford assay (Bio-Rad, Hercules, CA, USA). Lysates were separated by SDS-PAGE, transferred to polyvinylidene fluoride membrane filters, and analysed by western blotting. Detailed information on the antibodies used in this study is listed in Table 2. Uncropped and unedited blot images are shown in Supplementary Fig. 5.

**Cell growth assay.** Cells ($1 \times 10^4$) were seeded into a plastic tissue culture dish and cultured at 37 °C in a humidified $CO_2$ incubator for 5 days. The cells were counted periodically using a hemocytometer.

**RNA isolation, reverse transcription, and quantitative polymerase chain reaction.** Total RNA was isolated from cells or tumours using the RNeasy Plus Mini Kit (Qiagen) and subjected to reverse transcription (RT) using ReverTra Ace qPCR RT Master Mix (TOYOBO, Osaka, Japan). The RT products were then analysed by real-time PCR in a 7900HT qPCR system (Applied Biosystems; Foster City, CA, USA) using KOD SYBR qPCR (TOYOBO) and specific primers (Table 3) as previously described[28,58]. The expression levels of individual mRNAs were normalised to that of *ACTB* mRNA.

**Measurement of lactic acid and ATP.** Cells were seeded in 96-well plates ($2 \times 10^4$ and $5 \times 10^4$ cells per well for MDA-MB-231 cells and macrophages, respectively) in triplicate and cultured for 24 h with or without naphthofluorescein (10 µM). For lactic acid levels, conditioned medium cultured for 6 h in fresh medium was collected, and the lactic acid contents therein were measured using an L-Lactic Acid kit (R-Biopharm, Darmstadt, Germany). For the overexpression experiments, 50 ng/ well of pcDNA3 V5-tagged Mint3 or mock plasmid was transfected to MDA-MB-231 cells using Lipofectamine 3000 (Thermo Fischer Scientific) 2 h before compound treatment. ATP levels were determined using the ATP Bioluminescence Assay Kit CLS II (Roche Applied Science, Penzberg, Germany). The values obtained were normalised to protein concentrations that were determined using a Bradford Assay Kit (Bio-Rad), as previously described[18,19].

**Mice.** Mice were maintained under specific pathogen-free conditions. Experimental protocols were approved by the Animal Care and Use Committees of the Institute of Medical Science, University of Tokyo and Institute of Medical, Pharmaceutical and Health Sciences, Kanazawa University, and the experiments were conducted in accordance with the institutional ethical guidelines for animal experiments and the safety guidelines for gene manipulation experiments. The sample size is based on the statistical analysis of variance and on exploratory experiments. BALB/c nude mice and C57BL/6J mice were purchased from Clea Japan (Tokyo, Japan). Mint3 KO mice (C57BL/6J background) have been described previously[23] (Riken CLST Accession number: CDB0589K).

**Toxicity test of naphthofluorescein.** Eight-week-old male C57BL/6J mice were intraperitoneally injected with vehicle (saline containing 1% Cremophor EL [Sigma] and 8 mM $Na_2CO_3$) or naphthofluorescein (100 mg/kg b.w.) once daily for

5 consecutive days followed by 2 days off for 2 weeks, and the body weight of mice was periodically measured. On day 14, the mice were sacrificed, and sera, liver, kidneys, and heart were collected from each subject. Sera were subjected to biochemical tests by Oriental Yeast Co., Ltd. (Tokyo, Japan). The livers, kidneys, and hearts were fixed with 4% paraformaldehyde/PBS, dehydrated, embedded in paraffin, sliced at 4-µm thickness, and stained by ordinary hematoxylin and eosin staining prior to histological analysis.

**Tumour transplantation.** Six-week-old female C57BL/6J mice (E0771) and BALB/c nude mice (HT1080, MDA-MB-231, and AsPC-1) were used for the tumour growth assay using subcutaneous injection of tumour cells. For the growth studies, cells ($1 \times 10^6$ for E0771, HT1080, HT1080 expressing FLAG-tagged FIH-1, and MDA-MB-231; $2 \times 10^6$ for AsPC-1) in 0.2 mL PBS were injected into the left flank of mice. On day 7 or 10, mice were randomly assigned to each experimental group. For the immunoprecipitation analysis of FLAG-tagged FIH-1 expressing HT1080 tumours, mice were injected intraperitoneally with vehicle or naphthofluorescein (100 mg/kg b.w.), and tumours were collected 3 h after injection and subjected to immunoprecipitation. For the gene expression analysis of HT1080 tumours, mice were injected intraperitoneally with vehicle or naphthofluorescein (100 mg/kg b.w.), and tumours were collected 6 h after injection and subjected to RNA isolation. For the monotherapy experiment, vehicle or naphthofluorescein (100 mg/kg b.w.) was injected into mice once daily for five consecutive days followed by 2 days off treatment. For the combination therapy of naphthofluorescein and cytotoxic anticancer reagents, vehicle/naphthofluorescein (100 mg/kg b.w.) and vehicle/ anticancer reagents (doxorubicin 2 mg/kg b.w. for MDA-MB-231; gemcitabine 20 mg/kg b.w. for AsPC-1 tumours) were injected into tumour-bearing mice twice a week. Tumour volumes were periodically measured with a caliper using the formula:

$$V = \frac{4}{3} \times \pi \times \left(\frac{d}{2}\right)^2 \times \left(\frac{D}{2}\right), \tag{1}$$

where $d$ is the minor tumour axis and $D$ is the major tumour axis.

For the experimental lung metastasis assay, $4 \times 10^5$ B16F10 or LLC cells in 0.2 mL PBS were injected into 6-week-old male C57BL/6J mice via the lateral tail veins. The mice were sacrificed 14 days after injection. All metastatic foci on the lungs were counted under a stereoscopic microscope.

**Immunoprecipitation.** Tumours were lysed in RIPA buffer (Sigma) by sonication and centrifuged at $20,000 \times g$ for 30 min at 4 °C. The supernatants were collected, and the total protein content was measured using the Bradford assay (Bio-Rad). Lysates were subjected to immunoprecipitation using anti-FLAG M2 agarose beads (Sigma) as previously described[20].

**Flow cytometry.** Macrophages and lung cells were isolated as previously described[26]. Single-cell suspensions were prepared using 40-µm cell strainers (BD Biosciences) and treated with antibody cocktails on ice for 30 min. Detailed antibody information is listed in Supplementary Table 4. The cells were washed with PBS containing 2% FBS and sorted using a FACS Aria (BD Biosciences). Data were analysed using FlowJo software (Tree Star, Ashland, OR, USA). Flow cytometric analysis strategy for the definition of inflammatory monocytes is shown in Supplementary Fig. 6.

**Immunostaining.** Frozen sections of tumour tissues were prepared and subjected to immunostaining using specific antibodies (Supplementary Table 4) as previously described[26,58]. The nuclei were counterstained with Hoechst 33342, and the sections were observed by confocal microscopy (Olympus, Tokyo, Japan). Paraffin-embedded sections were prepared as previously described[28]. Sections were cut at a thickness of 4 µm, deparaffinized in xylene, and rehydrated in a decreasing graded ethanol series. Antigen retrieval was carried out for CD8α immunostaining by autoclaving in 10 mM citrate buffer (pH 9.0) for 10 min at 121 °C, followed by cooling for 20 min. After blocking endogenous peroxidase activity with a 3% aqueous $H_2O_2$ solution for 5 min, the sections were incubated with rabbit

**Table 2 List of antibodies.**

| Antigen | Host species | Company | Catalogue number | Dilution |
|---|---|---|---|---|
| β-actin | Mouse | FUJIFILM Wako | 0111-24554 | 1:5000 (WB) |
| Mint3 | Mouse | BD Biosciences | 611380 | 1:1000 (WB) |
| FIH-1 | Goat | SantaCruz | SC-26219 | 1:250 (WB) |
| His6 | Mouse | Roche | 11922416001 | 1:1000 (WB) |
| GST | Mouse | GeneTex | GTX70195 | 1:1000 (WB) |
| FLAG | Rabbit | Millipore | F7425 | 1:1000 (WB) |
| Gal4 | Mouse | SantaCruz | sc-510 | 1:1000 (WB) |
| MT1-MMP | Rabbit | Millipore | AB6004 | 1:1000 (WB) |
| Integrin α5 | Mouse | BD Biosciences | 51-9001996 | 1:1000 (WB) |
| CD8α | Rabbit | CST | 98941 | 1:200 (IHC) |
| E-selectin (CD62E) | Rat | Abcam | ab2497 | 1:200 (IFA) |
| Cleaved caspase-3 | Rabbit | CST | 9665 | 1:100 (IFA) |
| Furin | Rabbit | Abcam | ab3467 | 1:200 (IFA) |
| GM130 | Mouse | BD Biosciences | 610822 | 1:200 (IFA) |
| Rabbit IgG, HRP-linked whole Ab | Donkey | GE Healthcare | NA934V | 1:3000 (WB) |
| Mouse IgG, HRP-linked whole Ab | Sheep | GE Healthcare | NA931V | 1:3000 (WB) |
| Peroxidase-conjugated anti-goat IgG(H + L) | Rabbit | Proteintech | SA00001-4 | 1:5000 (WB) |
| Mouse IgG, Alexa Fluor 488 | Goat | Thermo Fisher Scientific | A-11029 | 1:2000 (IFA) |
| Rabbit IgG, Alexa Fluor 546 | Goat | Thermo Fisher Scientific | A-11035 | 1:2000 (IFA) |
| Rabbit IgG, Alexa Fluor 488 | Goat | Thermo Fisher Scientific | A-11034 | 1:500 (IFA) |
| Rat IgG, Alexa Fluor 594 | Goat | Thermo Fisher Scientific | A-11007 | 1:500 (IFA) |
| Mouse Ly-6G, FITC conjugate | Rat | TONBO Biosciences | 35-1276 | 1:400 (FACS) |
| Mouse MHC Class II(I-A/I-E), PE conjugate | Rat | TONBO Biosciences | 50-5321 | 1:200 (FACS) |
| Mouse Ly-6C, APC conjugate | Rat | eBioscience | 17-5932 | 1:200 (FACS) |
| Mouse/human CD11b, APC-Cy7 conjugate | Rat | BioLegend | 101226 | 1:200 (FACS) |
| Mouse CD45, PE-Cy7 conjugate | Rat | eBioscience | 25-0451-82 | 1:2000 (FACS) |
| Mouse CCR2, FITC conjugate | Rat | BioLegend | 150607 | 1:200 (FACS) |
| IgG2b, κ Isotype, FITC conjugate | Rat | BioLegend | 400605 | 1:200 (FACS) |

**Table 3 Primer pairs used in quantitative real-time polymerase chain reaction.**

| Gene | Direction | Sequence |
|---|---|---|
| ACTB | Forward (f) | 5'- TTCTACAATGAGCTGCGTGTG -3' |
| | Reverse (r) | 5'- GGGGTGTTGAAGGTCTCAAA -3' |
| SLC2A1 | f | GGGCATGTGCTTCCAGTATGT |
| | r | ACCAGGAGCACAGTGAAGAT |
| HK2 | f | GTCCACTCCAGATGGGACAG |
| | r | GGAGCCCATTGTCCGTTACT |
| PGK1 | f | GCATACCTGCTGGCTGGATG |
| | r | CCCACAGGACCATTCCACAC |
| PDK1 | f | TCCTGTCACCAGCCAGAATG |
| | r | CTTCCTTTGCCTTTTCCACC |
| LDHA | f | CTCCAAGCTGGTCATTATCACG |
| | r | AGTTCGGGCTGTATTTTACAACA |
| HIF1A | f | ATCCATGTGACCATGAGGAAATG |
| | r | CTCGGCTAGTTAGGGTACACTT |
| HIF1AN (FIH-1) | f | CCTCCGGATCAGTTCGAGTG |
| | r | TGTCAAAGTCCACCTGGCTC |
| 18S rRNA | f | GCTTAATTTGACTCAACACGGGA |
| | r | AGCTATCAATCTGTCAATCCTGTC |

monoclonal anti-mouse CD8α (1:200; # 98941; Cell Signaling Technology) at 4 °C overnight. After washing with Tris-buffered saline, CD8α antibodies were detected using the DAKO Envision+ Dual Link System (Agilent, Santa Clara, CA, USA). 3,3′-Diaminobenzidine tetrahydrochloride was used as a chromogen, whereas hematoxylin was used as a light counterstain. Immunostaining of the cells was performed using specific antibodies (Table 2) as previously described[58]. Cells were counterstained with Hoechst 33342, washed five times with PBS, mounted on slides, and imaged by fluorescent microscopy (BZ-X800; Keyence, Osaka, Japan).

**LPS-induced endotoxic shock model**. The LPS-induced endotoxic shock model was created as previously described, with some modifications[23]. Briefly, mice were injected intraperitoneally with vehicle or naphthofluorescein (100 mg/kg b.w.) 24 and 6 h before the intraperitoneal LPS (10 mg/kg b.w.) injection, and their survival was analysed.

**LPS-responsive cytokine production assay**. For in vivo analysis, serum was obtained at the indicated time after LPS injection and subjected to ELISA using ELISA kits for mouse TNF-α, IL-6, and IL-12p70 (R&D Systems). For in vitro analysis of cytokine production, macrophages ($2 \times 10^4$/well) were seeded into 96-well plates and stimulated with LPS (100 ng/mL) in the presence or absence of naphthofluorescein (10 μM) for 24 h. Supernatants of the macrophage cultures were collected, and cytokine levels were measured using ELISA kits.

**CCL2 ELISA**. CCL2 levels in the lungs and sera were analysed as previously described[26]. Briefly, mouse lungs were homogenised in lysis buffer (20 mM HEPES, 100 mM NaCl, 1% Nonidet P-40, 10% glycerol) containing Protease Inhibitor Cocktail Set II (1:100; Millipore). Lung lysates and sera were analysed with a mouse CCL2 ELISA kit (MJE00; R&D Systems).

**Statistics and reproducibility**. The Mann–Whitney $U$ test, two-sided unpaired Student's $t$ test with Welch's correction, and the chi-square test with Yates' correction were used for statistical evaluations using GraphPad Prism (GraphPad Software, Inc., La Jolla, CA, USA), as indicated in each experiment. For Kaplan–Meier analyses, the log-rank test with Benjamini–Hochberg adjustment was performed using the R packages "survival" and "survminer." Data are presented as the mean ± SEM or ±SD; $p$ values $< 0.05$ were considered significant. Sample size is based on statistical analysis of variance and on exploratory experiments. Each in vitro experiment was replicated at least three times successfully. Animal experiments were replicated at least twice successfully.

**Reporting summary**. Further information on research design is available in the Nature Research Reporting Summary linked to this article.

## Data availability

All of the data supporting the findings of this study are available in this article and in the supplementary information files (Supplementary Data 1).

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

## Acknowledgements

We thank Hiroki J. Nakaoka, Miho Ishiura, and Mai Nakayama for providing technical support. This work was supported by a Grant-in-Aid for Scientific Research (C) (19K07659) from MEXT; a P-CREATE (Project for Cancer Research and Therapeutic Evolution; grant number JP21cm0106211) grant from The Japan Agency for Medical Research and Development; and the Kobayashi Foundation for Cancer Research (to

T.S.); Grant-in-Aid for Scientific Research (C) (18K05366) and Grant-in-Aid for Scientific Research on Innovative Areas (18H05503) from MEXT (to Y.K.).

## Author contributions
T.Sakamoto and M.S. designed the study. T.Sakamoto, Y.K., and T.H. wrote the manuscript. T.S., Y.F., T.Hara., T.Hayashi, and Y.S. performed cell and mouse experiments. Y.K., K.H., T.Shimizu, and H.O. performed the chemical array screening. Y.M., J.I., S.K., and M.S. supervised the study. All authors have read and approved the final manuscript.

## Competing interests
The authors declare no competing interests.
