## [Peer Review File · Communications Biology]

Pharmacological inhibition of Mint3 attenuates tumour growth, metastasis, and endotoxic shockReviewers' comments:

Reviewer #1 (Remarks to the Author):

In the present manuscript the authors identified naphthofluorescein as a promising mint3 inhibitor, leading to disruption of the Mint3-FIH-1 interaction and thus attenuated HIF-1 activity. The number and quality of experiments demonstrating the target identification, way of action and functional validation in in vitro and in vivo models of acute, endotoxin-induced inflammation as well as cancer growth and metastasis is remarkable. In line with the suggested target of naphthofluorescein, namely the protein-protein interaction between Mint3 and FIH-1 leading to abrogation of the inhibitory effect of MINT3 to FIH-1, the presented results largely confirm similar experimental designs in mice lacking ABPA3/Mint3 expression, e.g. in macrophages. Thus, the potential of naphthofluorescein Derivatives as useful therapeutic candidate for cancer and inflammatory diseases can be clearly recognized.

HIF-1, macrophages also express HIF-2, and both transcription factors share common but not identical target genes. Since Mint3 inhibits FIH-1, Mint3 inhibition should also affect HIF-2. It is known that HIF-1-deficient macrophages exhibit a significant reduction in ATP production, whereas HIF-2-deficient macrophages do not exhibit any alteration in ATP levels. Hence, in line with the argumentation of the authors, the observed alterations in the glycolytic activity in macrophages should be primarily caused by the inhibition of HIF-1 activity. However could the authors comment, or show, if also HIF2 and in consequence HIF2 specific targets are altered in response to naphthofluorescein treatment. Otherwise the authors should better explain the selectivity of naphthofluorescein for HIF1 over HIF2 activation.

Fig. 5 and Fig. 6. The functions of tumor-associated, inflammatory macrophages include angiogenesis, suppression of antitumor immune responses, chemoresistance, and metastasis. For their recruitment from the bone marrow, the CCL2-CCR2 axis is a critical pathway. Have authors investigated an effect of naphthofluorescein treatment on the CCL2-CCR2 axis? Please comment.

Reviewer #2 (Remarks to the Author):

The manuscript entitled "Pharmacological inhibition of Mint3-FIH-1 interaction attenuates tumour growth, metastasis, and endotoxic shock in in vitro and in vivo analyses" describes a study to link the inhibition of the Mint3-FIH-1 interaction with the loss of HIF-1 activity as well as the downregulation of glycolytic enzymes in cancer cells and macrophages. Furthermore, in vivo experiments revealed a reduction in tumour growth after pharmacological inhibition of Mint3-FIH-1 interaction. In the study are interesting aspects, but in its current form the data require further validations.

Specifically, the authors should address the following points to improve the quality of the paper:

Specific comments:

- Introduction: Page 3, line 28-33. Rephrase the sentence.

Page 3, line 36: HIF-1 α knockout mice are embryonic lethal at day E11 due to cardiovascular malfunctions and neural tube defects.

Page 4, line 52: Specify "moderate hypoxic conditions"

- Throughout the manuscript only one siRNA/shRNA against Mint3 is used. It is important to show that 2 different siRNAs/shRNAs against the same protein have the same effect to minimize the off-target effects.

- Figure 1a: Specify which amino acids of the HIF-1 α protein were fused to the GalBD to measure HIF-1 α transactivation via luciferase activity or describe the plasmid more precisely in the material

and method part.

- Figure 1: The authors mentioned (page 5, line 77) that compound #9 showed severe toxicity to HT1080 cells. How was this measured? The authors should include cell viability assay or BrdU incorporation in figure 1.
- Figure 2 and 4: The authors should measure HIF-1 α mRNA levels as well, because one would suggest that treatment with naphthofluorescein has no effect on the HIF-1 α mRNA levels
- Figure 4b: The authors should provide the uncropped Western blot for Mint3 after Flag-IP (middle panel). In the photograph shown here the background on the right side is so much brighter than on the left side, which gives the impression that it is not the same Western blot.
- Figure 2, 3 and 4: Can the authors abolish the effect by overexpression of Mint3?
- Is this effect HIF-1 α specific? What about HIF-2? FIH can also hydroxylate HIF-2 α .

Reviewer #3 (Remarks to the Author):

In the manuscript COMMSBIO-21-0269-T, the authors report the compound naphthofluorescein as novel inhibitor of the Mint3-FIH-1 interaction and state that the inhibition of this interaction by naphthofluorescein attenuates tumor growth, metastasis formation and LPS-induced shock. However, the compound naphthofluorescein is a known furin inhibitor with a large array of effects, ranging from the regulation of cancer cell motility and invasiveness as well as blood brain barrier integrity to the inhibition of the furin-dependent cleavage of the SARS-CoV-2 spike protein (Coppola et al. *Neoplasia*. 2008;10:363-70; Baumann et al. *Exp Cell Res*. 2019;383:111503; Cheng et al. *Cell Rep*. 2020;33:108254). The authors fail to present convincing evidence that the in vivo effects of naphthofluorescein are specifically due to the inhibition of the Mint3-FIH-1 interaction. The authors show data that somewhat indicate naphthofluorescein to be a specific inhibitor for the Mint3-FIH-1 interaction using a single concentration in controlled cell culture experiments. However, comparable naphthofluorescein concentrations were used previously to effectively inhibit furin (Coppola et al. *Neoplasia*. 2008;10:363-70;). Moreover, in vivo naphthofluorescein tissue and cell concentrations are unknown and it is therefore not possible to translate such cell culture experiments to in vivo treatments.

The authors show that certain effects of naphthofluorescein treatment are lost following Mint3 knockout, but Mint3 can also effect furin localization (Han et al. *J Cell Sci*. 2008;121:2217-23) and Mint3 is hence not specifically regulating FIH. Furthermore, studies analysing the effect of HIF hydroxylase inhibitors on tumor growth and metastasis formation found that increased HIF activity is protective against both (Madsen et al. 2015 *EMBO Rep*. 2015;16:1394-408; Kachamakova-Trojanowska et al. *Biochem Pharmacol*. 2020;175:113922), in contrast to the assumptions of the authors made in this manuscript. Moreover, the HIF-1 target gene LDHA was shown to be regulated in its expression following naphthofluorescein treatment in this manuscript, but it was previously demonstrated that FIH does not affect the HIF-dependent regulation of LDHA transcription (Chan et al. *J Biol Chem*. 2016;291:20661-73).

Responses to reviewers

We really appreciate the reviewers for their valuable comments, to improve our manuscript. The added or modified sentences are written in red in the revised manuscript.

Reviewer #1:

In the present manuscript the authors identified naphthofluorescein as a promising mint3 inhibitor, leading to disruption of the Mint3–FIH-1 interaction and thus attenuated HIF-1 activity. The number and quality of experiments demonstrating the target identification, way of action and functional validation in *in vitro* and *in vivo* models of acute, endotoxin-induced inflammation as well as cancer growth and metastasis is remarkable. In line with the suggested target of naphthofluorescein, namely the protein-protein interaction between Mint3 and FIH-1 leading to abrogation of the inhibitory effect of MINT3 to FIH-1, the presented results largely confirm similar experimental designs in mice lacking ABPA3/Mint3 expression, e.g. in macrophages. Thus, the potential of naphthofluorescein Derivatives as useful therapeutic candidate for cancer and inflammatory diseases can be clearly recognized.

HIF-1, macrophages also express HIF-2, and both transcription factors share common but not identical target genes. Since Mint3 inhibits FIH-1, Mint3 inhibition should also affect HIF-2. It is known that HIF-1-deficient macrophages exhibit a significant reduction in ATP production, whereas HIF-2-deficient macrophages do not exhibit any alteration in ATP levels. Hence, in line with the argumentation of the authors, the observed alterations in the glycolytic activity in macrophages should be primarily caused by the inhibition of HIF-1 activity. However could the authors comment, or show, if also HIF2 and in consequence HIF2 specific targets are altered in response to naphthofluorescein treatment. Otherwise the authors should better explain the selectivity of naphthofluorescein for HIF1 over HIF2 activation.

Response:

FIH-1 has been reported to hydroxylate HIF-2 α much less effectively compared to HIF-1 α because of the difference in amino acid sequences around the target asparagine residue (Koivunen P et al., JBC, 2004; Bracken CP et al., JBC, 2006). We also confirmed that the overexpression of FIH-1 suppressed HIF-2 α reporter activity less effectively compared to HIF-1 α in HT1080 cells. Consistent with these results, naphthofluorescein treatment also showed little or no suppression of HIF-2 α reporter activity in HT1080 cells. Thus, naphthofluorescein shows a selectivity for HIF-1 rather than for HIF-2. We added the data on FIH-1 overexpression and naphthofluorescein treatment in luciferase assays in Supplementary Fig.

1e-g and explained these in the Results section, line 110, page 6 – line 116, page 7 in the revised manuscript.

Fig. 5 and Fig. 6. The functions of tumor-associated, inflammatory macrophages include angiogenesis, suppression of antitumor immune responses, chemoresistance, and metastasis. For their recruitment from the bone marrow, the CCL2–CCR2 axis is a critical pathway. Have authors investigated an effect of naphthofluorescein treatment on the CCL2–CCR2 axis? Please comment.

Response:

Mint3 depletion attenuates chemotaxis towards CCL2 in macrophages/inflammatory monocytes owing to the defect in ATP production via glycolysis without affecting the expression levels of CCR2 in macrophages and those of CCL2 in metastatic lungs and serum in tumour-inoculated mice (Hara T et al., JBC, 2011; Hara T et al., PNAS, 2017). We confirmed that naphthofluorescein did not affect the expression levels of CCR2 in macrophages and those of CCL2 in metastatic lungs and serum in tumour-inoculated mice. We added these data in Supplementary Fig. 4c-e and explained them in the Results section, lines 223-233, page 12 in the revised manuscript.

Reviewer #2:

The manuscript entitled “Pharmacological inhibition of Mint3-FIH-1 interaction attenuates tumour growth, metastasis, and endotoxic shock in in vitro and in vivo analyses” describes a study to link the inhibition of the Mint3-FIH-1 interaction with the loss of HIF-1 activity as well as the downregulation of glycolytic enzymes in cancer cells and macrophages. Furthermore, in vivo experiments revealed a reduction in tumour growth after pharmacological inhibition of Mint3-FIH-1 interaction. In the study are interesting aspects, but in its current form the data require further validations.

Specifically, the authors should address the following points to improve the quality of the paper:

Specific comments:

Introduction: Page 3, line 28-33. Rephrase the sentence.

Response: We rephrased the sentence in the revised manuscript as follows:

“HIF-1 α is negatively regulated by two types of hydroxylases, a HIF prolyl hydroxylase domain containing protein 1-3 (PHD1-3) and a factor inhibiting HIF-1 (FIH-1), in an oxygen-dependent manner. PHD1-3 hydroxylates two proline residues of HIF-1 α and thereby promotes proteasomal degradation of the HIF-1 α protein. In turn, FIH-1 hydroxylates an asparagine residue of HIF-1 α and thereby inactivates the transcriptional activity of HIF-1 α ” (Introduction, page 3, lines 28-33)

Page 3, line 36: HIF-1 α knockout mice are embryonic lethal at day E11 due to cardiovascular malfunctions and neural tube defects.

Response: We provided explanations on Mint3 KO mice and Mint3 conditional KO mice separately in the revised manuscript. (Introduction, page 3, lines 36-38)

Page 4, line 52: Specify “moderate hypoxic conditions”

Response: We rephrased the sentence as follows:

“Furthermore, the K_m values of FIH-1 and PHDs for O₂ are about 90 and 230 μ M, respectively, *in vitro* (Koivunen P et al., JBC, 2014), indicating that Mint3-dependent HIF-1 activation can be observed under normoxic and moderate hypoxic conditions where FIH-1 can hydroxylate HIF-1 α .” (Introduction, page 4, lines 52-55)

Throughout the manuscript only one siRNA/shRNA against Mint3 is used. It is important to show that 2 different siRNAs/shRNAs against the same protein have the same effect to minimize the off-target effects.

Response: We performed all knockdown experiments using two independent siRNA/shRNA with the same results and replaced the figures with the data obtained using two independent siRNA/shRNA.

Figure 1a: Specify which amino acids of the HIF-1 α protein were fused to the GalBD to measure HIF-1 α transactivation via luciferase activity or describe the plasmid more precisely in the material and method part.

Response: We used the plasmid expressing the 727-826 amino acid region of HIF-1 α fused to GalBD. We described this information in the Materials and Methods section, lines 378-379, page 18.

Figure 1: The authors mentioned (page 5, line 77) that compound #9 showed severe toxicity to HT1080 cells. How was this measured? The authors should include cell viability assay or BrdU incorporation in

figure 1.

Response:

We observed, by using a microscope, that almost all the cells treated with Compound #9 were cast off from the plate. Thus, we could not analyse the reporter activity of cells treated with Compound #9. We described this in the Results section, lines 78-80, page 5 in the revised manuscript.

Figure 2 and 4: The authors should measure HIF-1 α mRNA levels as well, because one would suggest that treatment with naphthofluorescein has no effect on the HIF-1 α mRNA levels

Response: We examined HIF-1 α mRNA levels in Naph-treated HT1080 cells and tumours of HT1080 cells from Naph-administrated mice. Naphthofluorescein did not affect HIF-1 α mRNA levels in both cases. We added these data in Fig. 2m and Fig. 4h and explained them in the Results section, line 120, page 7, and line 172, page 9 in the revised manuscript.

Figure 4b: The authors should provide the uncropped Western blot for Mint3 after Flag-IP (middle panel). In the photograph shown here the background on the right side is so much brighter than on the left side, which gives the impression that it is not the same Western blot.

Response: We uploaded the source data of all figures as Supplementary Table 6 including the full scans of the Western blot images.

Figure 2, 3 and 4: Can the authors abolish the effect by overexpression of Mint3?

Response: We examined whether Mint3 overexpression can abolish the effect of naphthofluorescein in HIF-1 reporter activity in HT1080 cells, lactate production in MDA-MB-231 cells, and HIF-1 target gene expression in tumours of HT1080 cells. Mint3 overexpression cancelled the effect of naphthofluorescein in these experiments. We added the data in Fig. 2d, e, Fig. 3c, d, and Supplementary Fig. 2c-g, and explained these in the Results section, lines 96-97, page 6; lines 140-141, page 8; and lines 172-174, page 9 in the revised manuscript.

Is this effect HIF-1 α specific? What about HIF-2? FIH can also hydroxylate HIF-2 α .

Response:

As described in the response to Reviewer#1, FIH-1 can hydroxylate HIF-2 α much less effectively than

HIF-1 α , and we confirmed that naphthofluorescein treatment showed little or no suppression of HIF-2 α reporter activity in HT1080 cells. Thus, naphthofluorescein shows the selectivity for HIF-1 rather than HIF-2.

Reviewer #3:

In the manuscript COMMSBIO-21-0269-T, the authors report the compound naphthofluorescein as novel inhibitor of the Mint3-FIH-1 interaction and state that the inhibition of this interaction by naphthofluorescein attenuates tumor growth, metastasis formation and LPS-induced shock. However, the compound naphthofluorescein is a known furin inhibitor with a large array of effects, ranging from the regulation of cancer cell motility and invasiveness as well as blood brain barrier integrity to the inhibition of the furin-dependent cleavage of the SARS-CoV-2 spike protein (Coppola et al. *Neoplasia*. 2008;10:363-70; Baumann et al. *Exp Cell Res*. 2019;383:111503; Cheng et al. *Cell Rep*. 2020;33:108254). The authors fail to present convincing evidence that the in vivo effects of naphthofluorescein are specifically due to the inhibition of the Mint3-FIH-1 interaction. The authors show data that somewhat indicate naphthofluorescein to be a specific inhibitor for the Mint3-FIH-1 interaction using a single concentration in controlled cell culture experiments. However, comparable naphthofluorescein concentrations were used previously to effectively inhibit furin (Coppola et al. *Neoplasia*. 2008;10:363-70;). Moreover, in vivo naphthofluorescein tissue and cell concentrations are unknown and it is therefore not possible to translate such cell culture experiments to in vivo treatments. The authors show that certain effects of naphthofluorescein treatment are lost following Mint3 knockout, but Mint3 can also effect furin localization (Han et al. *J Cell Sci*. 2008;121:2217-23) and Mint3 is hence not specifically regulating FIH.

Response:

Reviewer #3 has concerns that naphthofluorescein might affect furin activity/localisation and thereby show anti-tumour effects independently from the Mint3-FIH-1 axis, which we claimed. To address these concerns, we first examined whether naphthofluorescein administration affected the furin-mediated conversion of MT1-MMP from the pro-form to the mature form in tumours of FLAG-FIH-1 expressing HT1080 cells. Naphthofluorescein administration did not increase the pro-form of MT1-MMP in the tumours under the condition where an interaction between Mint3 and FLAG-tagged FIH-1 was inhibited by naphthofluorescein. Thus, naphthofluorescein inhibited Mint3-FIH-1 interaction without affecting furin activity in the tumours, at least under our experimental conditions. We added these data in Supplementary Fig. 2b and explained them in the Results section, lines 164-166, page 9 in the revised manuscript.

Next, we examined whether Mint3 knockdown and naphthofluorescein treatment affected the localisation of furin in the cells. Furin is mainly localised at the Golgi apparatus in HT1080 cells, and neither Mint3 knockdown nor naphthofluorescein treatment affected the localisation of furin in HT1080 cells. Thus, we could not reproduce the Mint3-mediated furin localisation at the Golgi apparatus, as reported by Han J et al., and naphthofluorescein did not also affect the localisation of furin in HT1080 cells. Furin is involved in various biological events by targeting various substrates including MT1-MMP, and furin knockout mice are embryonic lethal (Thomas G, *Nat Rev Mol Cell Biol*, 2002; Roebroek AJ et al., *Development*, 1998). In turn, Mint3 knockout mice showed no apparent abnormality (Ho et al., *J Neurosci*, 2006; Hara T et al., *JBC*, 2011; Chung Y et al., *BBR*, 2020). Naphthofluorescein administration did not also show acute toxicity in mice. Thus, Mint3/naphthofluorescein is not likely an essential regulator of furin, at least under the physiological conditions, although we do not exclude the possibility that furin inhibition partially contributed to the anti-tumour effects of naphthofluorescein, which become negligible when the Mint3-FIH-1 axis is disrupted. We added these data in Supplementary Fig. 1c, d, and explained them in the Results section, lines 106-108, page 6, and added these in the Discussion, in lines 271-279, page 14 in the revised manuscript.

To further confirm whether naphthofluorescein indeed targets the Mint3-FIH-1 axis, we examined the effects of naphthofluorescein on FIH-1 depleted cells in vitro and in vivo. In the reporter assay, naphthofluorescein decreased the HIF-1 reporter activity in control HT1080 cells but not in FIH-1-depleted cells. In the tumour growth assay, naphthofluorescein administration suppressed tumour growth in control MDA-MB-231 cells but not that in FIH-1-depleted cells. Thus, the effects of naphthofluorescein in vitro and in vivo depend on FIH-1. We added these data in the Results section, Fig. 2f, g, and Supplementary Fig. 3b-d, and explained them in lines 96-97, page 6 and lines 195-197, page 10 in the revised manuscript.

Furthermore, studies analysing the effect of HIF hydroxylase inhibitors on tumor growth and metastasis formation found that increased HIF activity is protective against both (Madsen et al. 2015 *EMBO Rep.* 2015;16:1394-408; Kachamakova-Trojanowska et al. *Biochem Pharmacol.* 2020;175:113922), in contrast to the assumptions of the authors made in this manuscript.

Response:

Many studies have revealed that HIF-1 inhibition is effective for cancer therapy in various experimental models, and our results of inhibition of Mint3-mediated HIF-1 activation via FIH-1 by naphthofluorescein are parallel to these previous reports (Semenza GL, *Oncogene*, 2010). In turn, PHD inhibitors have been

reported to show anti-tumour effects in some experimental models. PHD inhibitors induce accumulation of not only HIF-1 α but also HIF-2 α and systemic administration of PHD inhibitors increase erythropoiesis via HIF-2 accumulation in the kidney (Maxwell PH et al., Nat Rev Nephrol, 2016). This might improve hypoxic conditions in tumours resulting partly in the anti-tumour effects. In addition, HIF activities must be controlled properly for tumour progression in response to the tumour microenvironment. Thus, aberrant accumulation of HIFs caused by PHD inhibitors might also show anti-tumour effects similar to those of HIF inhibition.

We added these in the Discussion, in lines 298-307, page 15 in the revised manuscript.

Moreover, the HIF-1 target gene LDHA was shown to be regulated in its expression following naphthofluorescein treatment in this manuscript, but it was previously demonstrated that FIH does not affect the HIF-dependent regulation of LDHA transcription (Chan et al. J Biol Chem. 2016;291:20661-73).

It is difficult to compare the data obtained under different conditions; Chan et al analysed FIH-1-mediated LDHA expression in the presence of PHD inhibitors. In Fig. 4A of the paper by Chan et al, FIH-1 siRNA did not further increase LDHA mRNA levels in the presence of PHD inhibitors by microarray, though FIH-1 inhibitor increased FIH-1 mRNA levels in the presence of PHD inhibitors by qPCR in MCF-7 cells. FIH-1 inhibitor also increased LDHA mRNA levels moderately in the presence of PHD inhibitors in U2OS, Hep3B and HeLa cells in Fig. 5A, though the authors did not focus on these data. In addition, another paper by Dr. Ratcliffe's group shows that FIH-1 knockdown increased LDHA expression while FIH-1 overexpression decreased LDHA expression in U2OS cells (Stolze IP et al., JBC, 2004). Thus, FIH-1-mediated LDHA expression might be context dependent.

To clarify whether the suppression of LDHA and other HIF-1 target gene expression by naphthofluorescein depends on FIH-1, we analysed expression of these genes in control and FIH-1 knockdown HT1080 cells treated with DMSO or naphthofluorescein. Naphthofluorescein decreased expression of HIF-1 target genes including LDHA in control HT1080 cells but not in FIH-1 knockdown cells. Thus, suppression of HIF-1 target genes including LDHA by naphthofluorescein depends on FIH-1. We added these data in Supplementary Fig. 1h-m and explained them in the Results section, lines 121-123, page 7 in the revised manuscript.

Reviewers' comments:

Reviewer #1 (Remarks to the Author):

This manuscript has been significantly improved in revision. New experiments were conducted in response to specific critique. This body of work contains interesting and novel findings about the therapeutic potential of the Mint3–HIF-1 axis in the field of cancer and inflammatory diseases.

Reviewer #2 (Remarks to the Author):

The authors addressed the comments of the previous review and improved the quality of the paper. Thereby all of my concerns are clarified.

Reviewer #3 (Remarks to the Author):

In the revised version of this manuscript, the authors attempted to demonstrate a selectivity of naphthofluorescein for the interaction of MINT3 and furin. However, the performed experiments fail to convincingly show the suggested selectivity.

In Supplemental Figure 1C, D, the authors show two cells per experiment to claim that there was no difference in furin localization. This is not sufficient to be interpreted.

The authors attempted to show that furin was not inhibited in tumors treated with naphthofluorescein by analysing the cleavage status of MT1-MMP (Supplemental Figure 2B). The authors analysed homogenized tumor tissue by Western Blot and did not find a difference in the cleaved/non-cleaved form of MT1-MMP. However, MT1-MMP is also cleaved by other pro-protein convertases (Creemers and Khatib *Fron Biosci.* 2008;13:4960-71) and it cannot be excluded that these pro-protein convertases cleave MT1-MMP in the absence of furin activity. Thus, analysis of a single furin substrate (which is not specific for furin) is not sufficient to claim that furin was not inhibited by naphthofluorescein *in vivo*.

The authors further argue that furin deletion is embryonically lethal whereas naphthofluorescein treatment did not lead to a phenotype. However, naphthofluorescein was not given during embryonic development and induced furin deletion in adult liver does also not lead to a phenotype in healthy mice (Creemers and Khatib *Fron Biosci.* 2008;13:4960-71).

The authors show that FIH knockdown prevents the effect of naphthofluorescein on breast tumor growth. This is interesting and indicates that naphthofluorescein affects FIH activity *in vivo*. But such results would be necessary for all disease models in order to support the main claim of the authors.

Responses to reviewers

We appreciate the valuable comments from all reviewers, and their comments and suggestions have improved our manuscript. Added or modified sentences are indicated by red font in the revised manuscript.

Reviewer #1:

This manuscript has been significantly improved in revision. New experiments were conducted in response to specific critique. This body of work contains interesting and novel findings about the therapeutic potential of the Mint3–HIF-1 axis in the field of cancer and inflammatory diseases.

Response:

We very much appreciate your assuring comments.

Reviewer #2:

The authors addressed the comments of the previous review and improved the quality of the paper. Thereby all of my concerns are clarified.

Response:

We are deeply grateful for your favorable evaluation of our revised manuscript.

Reviewer #3

In the revised version of this manuscript, the authors attempted to demonstrate a selectivity of naphthofluorescein for the interaction of MINT3 and furin. However, the performed experiments fail to convincingly show the suggested selectivity.

In Supplemental Figure 1C, D, the authors show two cells per experiment to claim that there was no difference in furin localization. This is not sufficient to be interpreted.

Response:

We evaluated the furin localization in 50 cells per group. Using the chi-square test with Yates' correction, we analyzed whether the furin localization in the Golgi complex was affected by Mint3 knockdown or naphthofluorescein treatment. Most cells showed furin

localized to the Golgi network, and Mint3 knockdown, as well as naphthofluorescein treatment, did not significantly affect the localization of furin. A previous study (Han J et al., JCS, 2008) analyzed the localization of transiently overexpressed exogenous furin, whereas we analyzed the localization of endogenously expressed furin. This might be one of the reasons for different furin localizations in Mint3 knockdown cells between the previous study and our study. We added the results of the statistical analysis of furin localizations to Supplementary Fig. 1C and 1D.

The authors attempted to show that furin was not inhibited in tumors treated with naphthofluorescein by analysing the cleavage status of MT1-MMP (Supplemental Figure 2B). The authors analysed homogenized tumor tissue by Western Blot and did not find a difference in the cleaved/non-cleaved form of MT1-MMP. However, MT1-MMP is also cleaved by other pro-protein convertases (Creemers and Khatib Fron Biosci. 2008;13:4960-71) and it cannot be excluded that these pro-protein convertases cleave MT1-MMP in the absence of furin activity. Thus, analysis of a single furin substrate (which is not specific for furin) is not sufficient to claim that furin was not inhibited by naphthofluorescein *in vivo*.

Response:

We agree with the concerns expressed by the reviewer. Therefore, we analyzed another furin substrate, integrin $\alpha 5$, whose conversion from the pro-form is affected in liver-specific furin knockout mice (Roebroek AJ et al., JBC, 2004) in tumors of HT1080 cells. Naphthofluorescein administration did not affect the conversion of integrin $\alpha 5$ in tumors of HT1080 cells. These results support that naphthofluorescein administration does not affect furin activity in tumors of HT1080 cells. We added these data to Supplementary Fig. 2b. However, as the reviewer suggested, we cannot completely exclude the possibility that naphthofluorescein affects the conversion of some substrates by furin. Thus, we just explain the results in lines 165-168, page 9 as follows:

“We also confirmed that the administration of naphthofluorescein did not affect the conversion of the potential furin substrates MT1-MMP and integrin $\alpha 5$ from their pro-forms to the mature forms in these tumours (Supplementary Fig. 2b).”

Moreover, we added the possibility that direct and indirect furin suppression by Naph also contributes to the attenuation of tumor growth, metastasis, and endotoxic shock *in vivo* to the schematic diagram in Fig. 7h.

The authors further argue that furin deletion is embryonically lethal whereas naphthofluorescein treatment did not lead to a phenotype. However, naphthofluorescein was not given during embryonic development and induced furin deletion in adult liver does also not lead to a phenotype in healthy mice (Creemers and Khatib *Front Biosci.* 2008;13:4960-71).

Response:

We revised the sentences pointed out by the reviewer (page 14, lines 280-285 in the revised manuscript) as follows:

“Furin is involved in various biological events by targeting various substrates including MT1-MMP and integrin $\alpha 5$, and furin knockout in mice is embryonically lethal. In turn, Mint3 knockout mice show no apparent abnormality. Thus, Mint3 is not likely an essential regulator of furin, at least not during embryogenesis. However, we do not exclude the possibility that furin inhibition partially contributed to the anti-tumour effects of naphthofluorescein in Mint3-dependent and -independent manners (Fig. 7h).”

The authors show that FIH knockdown prevents the effect of naphthofluorescein on breast tumor growth. This is interesting and indicates that naphthofluorescein affects FIH activity *in vivo*. But such results would be necessary for all disease models in order to support the main claim of the authors.

Response:

Naphthofluorescein suppressed tumor growth of MDA-MB-231 cells in an FIH-1-dependent manner. In turn, naphthofluorescein suppressed host Mint3-mediated metastasis and LPS-induced endotoxic shock *in vivo*. However, as the reviewer suggested, we could not completely exclude the possibility that naphthofluorescein suppresses Mint3-mediated metastasis and endotoxic shock independently from FIH-1 *in vivo* in this study. Experiments using FIH-1 knockout or conditional knockout mice would be helpful to address this point. Because FIH-1 knockout mice show reduced body weight and hypermetabolism (Zhang N et al., *Cell Metab*, 2010), FIH-1 conditional knockout mice would be better for evaluating the specificity of naphthofluorescein on FIH-1-mediated effects. However, we do not have access to FIH-1 knockout/conditional knockout mice, and the generation of FIH-1 knockout mice

would have taken too much time. Thus, we discuss this limitation of our present study on page 14, lines 272-278, in the revised manuscript.

In addition, based on the reviewer's concerns, we have modified the title of the manuscript to "Pharmacological inhibition of Mint3 attenuates tumour growth, metastasis, and endotoxic shock" (the original title was "Pharmacological inhibition of Mint3-FIH-1 interaction attenuates tumour growth, metastasis, and endotoxic shock in *in vitro* and *in vivo* analyses") and carefully changed "the Mint3-FIH-1 axis" to "Mint3" in descriptions on metastasis and endotoxic shock *in vivo* models in the revised manuscript to avoid any overstatement.